# Deep Imitative Models for Flexible Inference, Planning, and Control

**Nicholas Rhinehart**
UC Berkeley
nrhinehart@berkeley.edu

**Rowan McAllister**
UC Berkeley
rmcallister@berkeley.edu

**Sergey Levine**
UC Berkeley
svlevine@berkeley.edu

## Abstract

Imitation Learning (IL) is an appealing approach to learn desirable autonomous behavior. However, *directing* IL to achieve arbitrary goals is difficult. In contrast, planning-based algorithms use dynamics models and reward functions to achieve goals. Yet, reward functions that evoke desirable behavior are often difficult to specify. In this paper, we propose "Imitative Models" to combine the benefits of IL and goal-directed planning. Imitative Models are probabilistic predictive models of desirable behavior able to plan interpretable expert-like trajectories to achieve specified goals. We derive families of flexible goal objectives, including constrained goal regions, unconstrained goal sets, and energy-based goals. We show that our method can use these objectives to successfully direct behavior. Our method substantially outperforms six IL approaches and a planning-based approach in a dynamic simulated autonomous driving task, and is efficiently learned from expert demonstrations without online data collection. We also show our approach is robust to poorly specified goals, such as goals on the wrong side of the road.

## 1 Introduction

Imitation learning (IL) is a framework for learning a model to mimic behavior. At test-time, the model pursues its best-guess of desirable behavior. By letting the model choose its own behavior, we cannot *direct* it to achieve different goals. While work has augmented IL with goal conditioning (Dosovitskiy & Koltun, 2016; Codevilla et al., 2018), it requires goals to be specified during training, explicit goal labels, and are simple (e.g., turning). In contrast, we seek flexibility to achieve *general* goals for which we have *no demonstrations*.

In contrast to IL, planning-based algorithms like model-based reinforcement learning (MBRL) methods do not require expert demonstrations. MBRL can adapt to new tasks specified through reward functions (Kuvayev & Sutton, 1996; Deisenroth & Rasmussen, 2011). The "model" is a dynamics model, used to *plan* under the user-supplied reward function. Planning enables these approaches to perform new tasks at test-time. The key drawback is that these models learn dynamics of *possible* behavior rather than dynamics of *desirable* behavior. This means that the responsibility of evoking desirable behavior is entirely deferred to engineering the input reward function. Designing reward functions that cause MBRL to evoke complex, desirable behavior is difficult when the space of possible *undesirable* behaviors is large. In order to succeed, the rewards cannot lead the model astray towards observations significantly different than those with which the model was trained.

Our goal is to devise an algorithm that combines the advantages of MBRL and IL by offering MBRL's flexibility to achieve new tasks at test-time and IL's potential to learn desirable behavior entirely from offline data. To accomplish this, we first train a model to forecast expert trajectories with a density function, which can *score trajectories and plans by how likely they are to come from the expert*. A probabilistic model is necessary because expert behavior is stochastic: e.g. at an intersection, the expert could choose to turn left or right. Next, we derive a principled probabilistic inference objective to create plans that incorporate both (1) the model and (2) arbitrary new tasks. Finally, we derive families of tasks that we can provide to the inference framework. Our method can accomplish *new tasks specified as complex goals* without having seen an expert complete these tasks before.

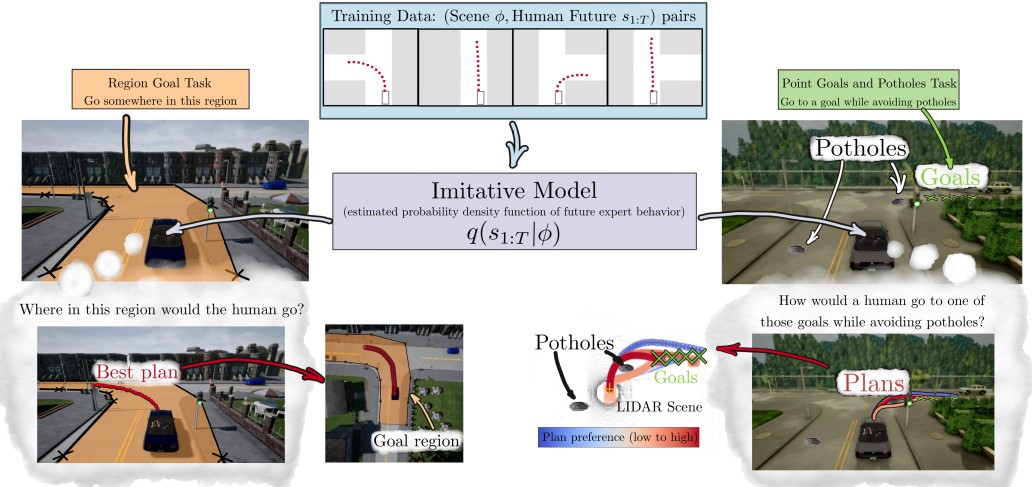

Figure 1: Our method: deep imitative models. *Top Center*. We use demonstrations to learn a probability density function $q$ of future behavior and deploy it to accomplish various tasks. *Left*: A region in the ground plane is input to a planning procedure that reasons about how the expert would achieve that task. It coarsely specifies a destination, and guides the vehicle to turn left. *Right*: Goal positions and potholes yield a plan that avoids potholes and achieves one of the goals on the right.

We investigate properties of our method on a dynamic simulated autonomous driving task (see Fig. 1). Videos are available at `https://sites.google.com/view/imitative-models`. Our contributions are as follows:

1. **Interpretable expert-like plans with minimal reward engineering.** Our method outputs multi-step expert-like plans, offering superior interpretability to one-step imitation learning models. In contrast to MBRL, our method generates expert-like behaviors with minimal reward engineering.

2. **Flexibility to new tasks:** In contrast to IL, our method flexibly incorporates and achieves goals not seen during training, and performs complex tasks that were never demonstrated, such as navigating to goal regions and avoiding test-time only potholes, as depicted in Fig. 1.

3. **Robustness to goal specification noise:** We show that our method is robust to noise in the goal specification. In our application, we show that our agent can receive goals on the wrong side of the road, yet still navigate towards them while staying on the correct side of the road.

4. **State-of-the-art CARLA performance:** Our method substantially outperforms MBRL, a custom IL method, and all five prior CARLA IL methods known to us. It learned near-perfect driving through dynamic and static CARLA environments from expert observations alone.

## 2 DEEP IMITATIVE MODELS

We begin by formalizing assumptions and notation. We model continuous-state, discrete-time, Partially-Observed Markov Decision Processes (POMDPs). For brevity, we call the components of state of which we have direct observations the agent's "state", although we explicitly assume these states do not represent the full Markovian world state. Our agent's state at time $t$ is $\mathbf{s}_t \in \mathbb{R}^D$; $t = 0$ refers to the current time step, and $\phi$ is *all* of the agent's observations. Variables are bolded. Random variables are capitalized. Absent subscripts denote *all* future time steps, e.g. $\mathbf{S} \doteq \mathbf{S}_{1:T} \in \mathbb{R}^{T \times D}$. We denote a probability density function of a random variable $\mathbf{S}$ as $p(\mathbf{S})$, and its value as $p(\mathbf{s}) \doteq p(\mathbf{S} = \mathbf{s})$.

To learn agent dynamics that are possible and preferred, we construct a model of expert behavior. We fit an "Imitative Model" $q(\mathbf{S}_{1:T}|\phi) = \prod_{t=1}^{T} q(\mathbf{S}_t|\mathbf{S}_{1:t-1}, \phi)$ to a dataset of expert trajectories $\mathcal{D} = \{(s^i, \phi^i)\}_{i=1}^{N}$ drawn from a (unknown) distribution of expert behavior $s^i \sim p(\mathbf{S}|\phi^i)$. By training $q(\mathbf{S}|\phi)$ to forecast expert trajectories with high likelihood, we model the scene-conditioned expert dynamics, which can score trajectories by how likely they are to come from the expert.

## 2.1 Imitative Planning to Goals

After training, $q(\mathbf{S}|\phi)$ can generate trajectories that resemble those that the expert might generate – e.g. trajectories that navigate roads with expert-like maneuvers. However, these maneuvers will not have a specific goal. Beyond generating human-like behaviors, we wish to *direct* our agent to goals and have the agent automatically reason about the necessary mid-level details. We define general tasks by a set of goal variables $\mathcal{G}$. The probability of a plan $\mathbf{s}$ conditioned on the goal $\mathcal{G}$ is modelled by a posterior $p(\mathbf{s}|\mathcal{G}, \phi)$. This posterior is implemented with $q(\mathbf{s}|\phi)$ as a *learned* imitation prior and $p(\mathcal{G}|\mathbf{s}, \phi)$ as a *test-time* goal likelihood. We give examples of $p(\mathcal{G}|\mathbf{s}, \phi)$ after deriving a maximum a posteriori inference procedure to generate expert-like plans that achieve abstract goals:

$$\mathbf{s}^* \doteq \arg\max_{\mathbf{s}} \; \log p(\mathbf{s}|\mathcal{G}, \phi) \;=\; \arg\max_{\mathbf{s}} \; \log q(\mathbf{s}|\phi) + \log p(\mathcal{G}|\mathbf{s}, \phi) - \log p(\mathcal{G}|\phi)$$
$$= \arg\max_{\mathbf{s}} \; \log \underbrace{q(\mathbf{s}|\phi)}_{\text{imitation prior}} + \log \underbrace{p(\mathcal{G}|\mathbf{s}, \phi)}_{\text{goal likelihood}}. \tag{1}$$

We perform gradient-based optimization of Eq. 1, and defer this discussion to Appendix A. Next, we discuss several goal likelihoods, which direct the planning in different ways. They communicate *goals* they desire the agent to achieve, but not how to achieve them. The planning procedure determines how to achieve them by producing paths similar to those an expert would have taken to reach the given goal. In contrast to black-box one-step IL that predicts controls, our method produces interpretable *multi-step plans* accompanied by two scores. One estimates the plan's "expertness", the second estimates its probability to achieve the goal. Their sum communicates the plan's overall quality.

Our approach can also be viewed as a learning-based method to integrate mid-level and high-level controllers together, where demonstrations from both are available at train-time, only the high-level controller is available at test-time, and the high-level controller can *vary*. The high-level controller's action specifies a subgoal for the mid-level controller. A density model of future trajectories of an expert mid-level controller is learned at train-time, and is amenable to different types of direction as specified by the high-level controller. In this sense, the model is an "apprentice", having learned to imitate mid-level behaviors. In our application, the high-level controller is composed of an $A^*$ path-planning algorithm and one of a library of components that forms goal likelihoods from the waypoints produced by $A^*$. Connecting this to related approaches, learning the mid-level controller (Imitative Model) resembles offline IL, whereas inference with an Imitative Model resembles trajectory optimization in MBRL, given goals provided by the high-level controller.

## 2.2 Constructing Goal Likelihoods

**Constraint-based planning to goal sets (hyperparameter-free)**: Consider the setting where we have access to a set of desired final states, one of which the agent should achieve. We can model this by applying a Dirac-delta distribution on the final state, to ensure it lands in a goal set $\mathbb{G} \subset \mathbb{R}^D$:

$$p(\mathcal{G}|\mathbf{s}, \phi) \leftarrow \delta_{\mathbf{s}_T}(\mathbb{G}), \quad \delta_{\mathbf{s}_T}(\mathbb{G}) = 1 \text{ if } \mathbf{s}_T \in \mathbb{G}, \quad \delta_{\mathbf{s}_T}(\mathbb{G}) = 0 \text{ if } \mathbf{s}_T \notin \mathbb{G}. \tag{2}$$

$\delta_{\mathbf{s}_T}(\mathbb{G})$'s *partial support* of $\mathbf{s}_T \in \mathbb{G} \subset \mathbb{R}^D$ constrains $\mathbf{s}_T$ and introduces *no hyperparameters* into $p(\mathcal{G}|\mathbf{s}, \phi)$. For each choice of $\mathbb{G}$, we have a different way to provide high-level task information to the agent. The simplest choice for $\mathbb{G}$ is a *finite* set of points: a **(A) Final-State Indicator** likelihood. We applied (A) to a sequence of *waypoints* received from a standard $A^*$ planner (provided by the CARLA simulator), and outperformed all prior dynamic-world CARLA methods known to us. We can also consider providing an *infinite* number of points. Providing a set of line-segments as $\mathbb{G}$ yields a **(B) Line-Segment Final-State Indicator** likelihood, which encourages the final state to land along one of the segments. Finally, consider a **(C) Region Final-State Indicator** likelihood in which $\mathbb{G}$ is a polygon (see Figs. 1 and 4). Solving Eq. 1 with (C) amounts to *planning the most expert-like trajectory that ends inside a goal region*. Appendix B provides derivations, implementation details, and additional visualizations. We found these methods to work well when $\mathbb{G}$ contains "expert-like" final position(s), as the prior strongly penalizes plans ending in non-expert-like positions.

**Unconstrained planning to goal sets (hyperparameter-based)**: Instead of *constraining* that the final state of the trajectory reach a goal, we can use a goal likelihood with *full support* ($\mathbf{s}_T \in \mathbb{R}^D$), centered at a desired final state. *This lets the goal likelihood encourage goals, rather than dictate them.* If there is a single desired goal ($\mathbb{G} = \{\mathbf{g}_T\}$), the **(D) Gaussian Final-State** likelihood $p(\mathcal{G}|\mathbf{s}, \phi) \leftarrow$

$\mathcal{N}(\mathbf{g}_T; \mathbf{s}_T, \epsilon I)$ treats $\mathbf{g}_T$ as a *noisy* observation of a final future state, and encourages the plan to arrive at a final state. We can also plan to $K$ successive states $\mathcal{G} = (\mathbf{g}_{T-K+1}, \ldots, \mathbf{g}_T)$ with a **(E) Gaussian State Sequence:** $p(\mathcal{G}|\mathbf{s}, \phi) \leftarrow \prod_{k=T-K+1}^{T} \mathcal{N}(\mathbf{g}_k; \mathbf{s}_k, \epsilon I)$ if a program wishes to specify a desired end velocity or acceleration when reaching the final state $\mathbf{g}_T$ (Fig. 2). Alternatively, a planner may propose a set of states with the intention that the agent should reach any one of them. This is possible by using a **(F) Gaussian Final-State Mixture:** $p(\mathcal{G}|\mathbf{s}, \phi) \leftarrow \frac{1}{K} \sum_{k=1}^{K} \mathcal{N}(\mathbf{g}_T^k; \mathbf{s}_T, \epsilon I)$ and is useful if some of those final states are not reachable with an expert-like plan. Unlike A–C, D–F introduce a hyperparameter "$\epsilon$". However, they are useful when *no states in* $\mathbb{G}$ *correspond to observed expert behavior*, as they allow the imitation prior to be robust to poorly specified goals.

**Costed planning**: Our model has the additional flexibility to accept arbitrary user-specified costs $c$ at test-time. For example, we may have updated knowledge of new hazards at test-time, such as a given map of potholes or a predicted cost map. Cost-based knowledge $c(\mathbf{s}_i|\phi)$ can be incorporated as an **(G) Energy-based** likelihood: $p(\mathcal{G}|\mathbf{s}, \phi) \propto \prod_{t=1}^{T} e^{-c(\mathbf{s}_t|\phi)}$ (Todorov, 2007; Levine, 2018). This can be combined with other goal-seeking objectives by simply *multiplying* the likelihoods together. Examples of combining G (energy-based) with F (Gaussian mixture) were shown in Fig. 1 and are shown in Fig. 3. Next, we describe instantiating $q(\mathbf{S}|\phi)$ in CARLA (Dosovitskiy et al., 2017).

Designing general goal likelihoods can be considered a form of reward engineering if there are no restrictions on the goal likelihoods. This connection is best seen in (G), which has an explicit cost term. One reason why it is easier to design goal likelihoods than to design reward functions is that the task of evoking most aspects of goal-driven behavior is already learned by the prior $q(\mathbf{s}|\phi)$, which models desirable behavior. This is in contrast to model-free RL, which entirely relies on the reward design to evoke goal-driven behavior, and in contrast to model-based RL, which heavily relies on the reward design to evoke goal-driven behavior, as its dynamics model learns what is possible, rather than what is desirable. Additionally, it is easy to design goal likelihoods when goals provide a significant amount of information that obviates the need to do any manual tuning. The main assumption is that one of the goals in the goal set is reachable within the model's time-horizon.

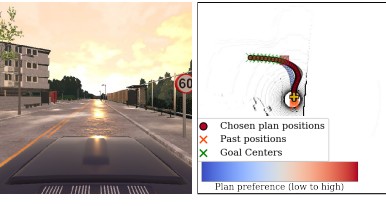 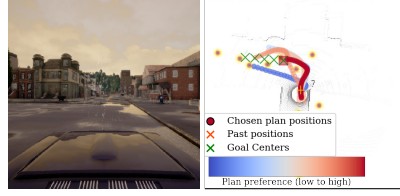 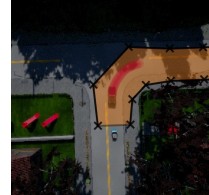

Figure 2: Imitative planning with the Gaussian State Sequence enables fine-grained control of the plans.

Figure 3: Costs can be assigned to "pot-holes" only seen at test-time. The planner prefers routes avoiding potholes.

Figure 4: Goal regions can be coarsely specified to give directions.

## 2.3 Applying Deep Imitative Models to Autonomous Driving

In our autonomous driving application, we model the agent's state at time $t$ as $\mathbf{s}_t \in \mathbb{R}^D$ with $D = 2$; $\mathbf{s}_t$ represents our agent's location on the ground plane. The agent has access to environment perception $\phi \leftarrow \{\mathbf{s}_{-\tau:0}, \boldsymbol{\chi}, \boldsymbol{\lambda}\}$, where $\tau$ is the number of past positions we condition on, $\boldsymbol{\chi}$ is a high-dimensional observation of the scene, and $\boldsymbol{\lambda}$ is a low-dimensional traffic light signal. $\boldsymbol{\chi}$ could represent either LIDAR or camera images (or both), and is the agent's observation of the world. In our setting, we featurize LIDAR to $\boldsymbol{\chi} = \mathbb{R}^{200 \times 200 \times 2}$, with $\boldsymbol{\chi}_{ij}$ representing a 2-bin histogram of points above and at ground level in a $0.5\text{m}^2$ cell at position $(i, j)$. CARLA provides ground-truth $\mathbf{s}_{-\tau:0}$ and $\boldsymbol{\lambda}$. Their availability is a realistic input assumption in perception-based autonomous driving pipelines.

**Model requirements:** A deep imitative model forecasts future expert behavior. It must be able to compute $q(\mathbf{s}|\phi) \forall \mathbf{s} \in \mathbb{R}^{T \times D}$. The ability to compute $\nabla_{\mathbf{s}} q(\mathbf{s}|\phi)$ enables gradient-based optimization for planning. Rudenko et al. (2019) provide a recent survey on forecasting agent behavior. As many forecasting methods cannot compute trajectory probabilities, we must be judicious in choosing $q(\mathbf{S}|\phi)$. A model that can compute probabilities R2P2 (Rhinehart et al., 2018), a generative autoregressive flow (Rezende & Mohamed, 2015; Oord et al., 2017). We extend R2P2 to instantiate the deep imitative

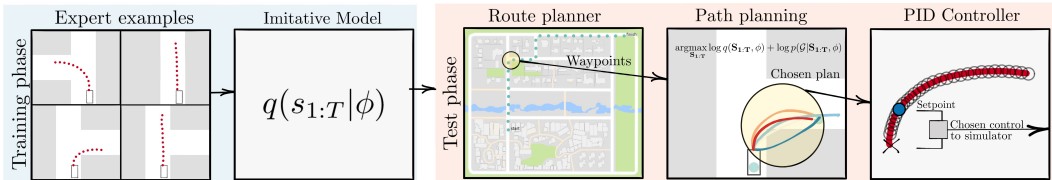

Figure 5: Illustration of our method applied to autonomous driving. Our method trains an imitative model from a dataset of expert examples. After training, the model is repurposed as an imitative planner. At test-time, a route planner provides waypoints to the imitative planner, which computes expert-like paths to each goal. The best plan is chosen according to the planning objective and provided to a low-level PID-controller in order to produce steering and throttle actions. This procedure is also described with pseudocode in Appendix A.

model $q(\mathbf{S}|\phi)$. R2P2 was previously used to forecast vehicle trajectories: it was not demonstrated or developed to plan or execute controls. Although we used R2P2, other future-trajectory density estimation techniques could be used – designing $q(\mathbf{s}|\phi)$ is not the primary focus of this work. In R2P2, $q_\theta(\mathbf{S}|\phi)$ is induced by an invertible, differentiable function: $\mathbf{S} = f_\theta(\mathbf{Z};\phi): \mathbb{R}^{T\times2} \mapsto \mathbb{R}^{T\times2}$; $f_\theta$ warps a latent sample from a base distribution $\mathbf{Z} \sim q_0 = \mathcal{N}(0, I)$ to $\mathbf{S}$. $\theta$ is trained to maximize $q_\theta(\mathbf{S}|\phi)$ of expert trajectories. $f_\theta$ is defined for $1..T$ as follows:

$$\mathbf{S}_t = f_t(\mathbf{Z}_{1:t}) = \mu_\theta(\mathbf{S}_{1:t-1}, \phi) + \sigma_\theta(\mathbf{S}_{1:t-1}, \phi)\mathbf{Z}_t, \qquad (3)$$

where $\mu_\theta(\mathbf{S}_{1:t-1}, \phi) = \mathbf{S}_{t-1} + (\mathbf{S}_{t-1} - \mathbf{S}_{t-2}) + m_\theta(\mathbf{S}_{1:t-1}, \phi) = 2\mathbf{S}_{t-1} - \mathbf{S}_{t-2} + m_\theta(\mathbf{S}_{1:t-1}, \phi)$ encodes a constant-velocity inductive bias. The $m_\theta \in \mathbb{R}^2$ and $\sigma_\theta \in \mathbb{R}^{2\times2}$ are computed by expressive neural networks. The resulting trajectory distribution is complex and multimodal (Appendix C.1 depicts samples). Because traffic light state was not included in the $\phi$ of R2P2's "RNN" model, it could not react to traffic lights. We created a new model that includes $\boldsymbol{\lambda}$. It fixed cases where $q(\mathbf{S}|\phi)$ exhibited no forward-moving preference when the agent was already stopped, and improved $q(\mathbf{S}|\phi)$'s stopping preference at red lights. We used $T = 40$ trajectories at 10Hz (4 seconds), and $\tau = 3$. Fig. 12 in Appendix C depicts the architecture of $\mu_\theta$ and $\sigma_\theta$.

## 2.4 IMITATIVE DRIVING

We now instantiate a complete autonomous driving framework based on imitative models to study in our experiments, seen in Fig. 5. We use three layers of spatial abstraction to plan to a faraway destination, common to autonomous vehicle setups: coarse route planning over a road map, path planning within the observable space, and feedback control to follow the planned path (Paden et al., 2016; Schwarting et al., 2018). For instance, a route planner based on a conventional GPS-based navigation system might output waypoints roughly in the lanes of the desired direction of travel, but not accounting for environmental factors such as the positions of other vehicles. This roughly communicates *possibilities* of where the vehicle could go, but not *when* or *how* it could get to them, or any environmental factors like other vehicles. A goal likelihood from Sec. 2.2 is formed from the route and passed to the planner, which generates a state-space plan according to the optimization in Eq. 1. The resulting plan is fed to a simple PID controller on steering, throttle, and braking. Pseudocode of the driving and inference algorithms are given in Algs 1 and 2. The PID algorithm is given in Appendix A.

## 3 RELATED WORK

A body of previous work has explored offline IL (Behavior Cloning – BC) in the CARLA simulator (Li et al., 2018; Liang et al., 2018; Sauer et al., 2018; Codevilla et al., 2018; 2019). These BC approaches condition on goals drawn from a small discrete set of directives. Despite BC's theoretical drift shortcomings (Ross et al., 2011), these methods still perform empirically well. *These approaches and ours share the same high-level routing algorithm: an A\* planner on route nodes that generates waypoints*. In contrast to our approach, these approaches use the waypoints in a *Waypoint Classifier*, which reasons about the map and the geometry of the route to classify the waypoints into one of several directives: {Turn left, Turn right, Follow Lane, Go Straight}. One of the original motivations for

---

**Algorithm 1** IMITATIVEDRIVING(ROUTEPLAN, IMITATIVEPLAN, PIDCONTROLLER, $q_\theta, f, p, H$)

---

1: $\phi \leftarrow$ ENVIRONMENT($\varnothing$) {Initialize the robot}
2: **while** not at destination **do**
3:    $\mathcal{G} \leftarrow$ ROUTEPLAN($\phi$) {Generate goals from a route}
4:    $\mathbf{s}_{1:T}^{\mathcal{G}} \leftarrow$ IMITATIVEPLAN($q_\theta, f, p, \mathcal{G}, \phi$) {Plan path}
5:    **for** $h = 0$ to $H$ **do**
6:      $u \leftarrow$ PIDCONTROLLER($\phi, \mathbf{s}_{1:T}^{\mathcal{G}}, h, H$)
7:      $\phi \leftarrow$ ENVIRONMENT($u$) {Execute control}
8:    **end for**
9: **end while**

---

**Algorithm 2** IMITATIVEPLAN($q_\theta, f, p, \mathcal{G}, \phi$)

---

1: Initialize $\mathbf{z}_{1:T} \sim q_0$
2: **while** not converged **do**
3:    $\mathbf{z}_{1:T} \leftarrow \mathbf{z}_{1:T} + \nabla_{\mathbf{z}_{1:T}} [\log q(f(\mathbf{z}_{1:T})|\phi) + \log p(\mathcal{G}|f(\mathbf{z}_{1:T}), \phi)]$ {Gradient ascent on Eq. 1}
4: **end while**
5: **return** $\mathbf{s}_{1:T} = f(\mathbf{z}_{1:T})$

---

these type of controls was to enable *a human* to direct the robot (Codevilla et al., 2018). However, in scenarios where there is no human in the loop (i.e. autonomous driving), we advocate for approaches to make use of the detailed spatial information inherent in these waypoints. Our approach and several others we designed make use of this spatial information. One of these is CIL-States (CILS): whereas the approach in Codevilla et al. (2018) uses images to directly generate controls, CILS uses identical inputs and PID controllers as our method. With respect to prior conditional IL methods, our main approach has more flexibility to handle more complex directives post-training, the ability to learn without goal labels, and the ability to generate interpretable planned and unplanned trajectories. These contrasting capabilities are illustrated in Table 1.

Our approach is also related to MBRL. MBRL can also plan with a predictive model, but its model only represents *possible* dynamics. The task of evoking expert-like behavior is offloaded to the reward function, which can be difficult and time-consuming to craft properly. We know of no MBRL approach previously applied to CARLA, so we devised one for comparison. *This MBRL approach also uses identical inputs to our method*, instead to plan a reachability tree (LaValle, 2006) over an dynamic obstacle-based reward function. See Appendix D for further details of the MBRL and CILS methods, which we emphasize use the *same inputs* as our method.

Several prior works (Tamar et al., 2016; Amos et al., 2018; Srinivas et al., 2018) used imitation learning to train policies that contain planning-like modules as part of the model architecture. While our work also combines planning and imitation learning, ours captures a distribution over possible trajectories, and then plan trajectories at test-time that accomplish a variety of given goals with high probability under this distribution. Our approach is suited to offline-learning settings where interactively collecting data is costly (time-consuming or dangerous). However, there exists online IL approaches that seek to be *safe* (Menda et al., 2017; Sun et al., 2018; Zhang & Cho, 2017).

## 4 EXPERIMENTS

We evaluate our method using the CARLA driving simulator (Dosovitskiy et al., 2017). We seek to answer four primary questions: **(1) Can we generate interpretable, expert-like plans with offline learning and minimal reward engineering**? Neither IL nor MBRL can do so. It is straightforward to *interpret* the trajectories by visualizing them on the ground plane; we thus seek to validate whether these plans are *expert-like* by equating expert-like behavior with high performance on the CARLA benchmark. **(2) Can we achieve state-of-the-art CARLA performance using resources commonly available in real autonomous vehicle settings?** There are several differences between the approaches, as discussed in Sec 3 and shown in Tables 1 and 2. Our approach uses the CARLA toolkit's resources that are commonly available in real autonomous vehicle settings: waypoint-based routes (all prior approaches use these), LIDAR and traffic-light observations (both are CARLA-

Table 1: Desirable attributes of each approach. A green check denotes that a method has a desirable attribute, whereas a red cross denotes the opposite. A "†" indicates an approach we implemented.

| Approach | Flexible to New Goals | Trains without goal labels | Outputs Plans | Trains Offline | Has Expert P.D.F. |
|---|---|---|---|---|---|
| CIRL* (Liang et al., 2018) | ✗ | ✗ | ✗ | ✗ | ✗ |
| CAL* (Sauer et al., 2018) | ✗ | ✗ | ✗ | ✓ | ✗ |
| MT* (Li et al., 2018) | ✗ | ✗ | ✗ | ✓ | ✗ |
| CIL* (Codevilla et al., 2018) | ✗ | ✗ | ✗ | ✓ | ✗ |
| CILRS* (Codevilla et al., 2019) | ✗ | ✗ | ✗ | ✓ | ✗ |
| CILS† | ✗ | ✓ | ✗ | ✓ | ✗ |
| MBRL† | ✓ | ✓ | ✓ | ✗ | ✗ |
| Imitative Models (*Ours*)† | ✓ | ✓ | ✓ | ✓ | ✓ |

Table 2: Algorithmic components of each approach. A "†" indicates an approach we implemented.

| Approach | Control Algorithm | ← Learning Algorithm | ← Goal-Generation Algorithm | ← Routing Algorithm | High-Dim. Obs. |
|---|---|---|---|---|---|
| CIRL* (Liang et al., 2018) | Policy | Behavior Cloning+RL | Waypoint Classifier | A* Waypointer | Image |
| CAL* (Sauer et al., 2018) | PID | Affordance Learning | Waypoint Classifier | A* Waypointer | Image |
| MT* (Li et al., 2018) | Policy | Behavior Cloning | Waypoint Classifier | A* Waypointer | Image |
| CIL* (Codevilla et al., 2018) | Policy | Behavior Cloning | Waypoint Classifier | A* Waypointer | Image |
| CILRS* (Codevilla et al., 2019) | Policy | Behavior Cloning | Waypoint Classifier | A* Waypointer | Image |
| CILS† | PID | Trajectory Regressor | Waypoint Classifier | A* Waypointer | (LIDAR,$\lambda$) |
| MBRL† | Reachability Tree | State Regressor | Waypoint Selector | A* Waypointer | (LIDAR,$\lambda$) |
| Imitative Models (*Ours*)† | Imitative Plan+PID | Traj. Density Est. | Goal Likelihoods | A* Waypointer | (LIDAR,$\lambda$) |

provided, but only the approaches we implemented use it). Furthermore, the two additional methods of comparison we implemented (CILS and MBRL) use the exact same inputs as our algorithm. These reasons justify an overall performance comparison to answer (2): whether we can achieve state-of-the-art performance using commonly available resources. We advocate that other approaches also make use of such resources. **(3) How flexible is our approach to new tasks?** We investigate (3) by applying each of the goal likelihoods we derived and observing the resulting performance. **(4) How robust is our approach to error in the provided goals?** We do so by injecting two different types of error into the waypoints and observing the resulting performance.

We begin by training $q(\mathbf{S}|\phi)$ on a dataset of 25 hours of driving we collected in Town01, detailed in Appendix C.2. Following existing protocol, each test episode begins with the vehicle starting in one of a finite set of starting positions provided by the CARLA simulator in Town01 or Town02 maps in one of two settings: static-world (no other vehicles) or dynamic-world (with other vehicles). We ran the same benchmark 3 times across different random seeds to quantify means and their standard errors. We construct the goal set $\mathbb{G}$ for the Final-State Indicator (A) directly from the route output by CARLA's waypointer. B's line segments are formed by connecting the waypoints to form a piecewise linear set of segments. C's regions are created a *polygonal goal region* around the segments of (B). Each represents an increasing level of coarseness of direction. Coarser directions are easier to specify when there is ambiguity in positions (both the position of the vehicle and the position of the goals). Further details are discussed in Appendix B.3. Visualizations of (C) are shown in Figures 6 and 7. Visualizations of (A) and (B) are shown in Figures 8 and 9. We use three metrics: (a) success rate in driving to the destination without any collisions (which all prior work reports); (b) red-light violations; and (c) proportion of time spent driving in the wrong lane and off road. With the exception of metric (a), lower numbers are better.

**Results**: Towards questions (1) and (3) (expert-like plans and flexibility), we apply our approach with a variety of goal likelihoods to the CARLA simulator. Towards question (2), we compare our methods against CILS, MBRL, and prior work. These results are shown in Table 3. The metrics for the methods we did not implement are from the aggregation reported in Codevilla et al. (2019). We observe our method to outperform all other approaches in all settings: static world, dynamic world, training conditions, and test conditions. We observe the *Goal Indicator methods are able to perform well, despite having no hyperparameters to tune*. We found that we could further improve our approach's performance if we use the light state to define different goal sets, which defines a "smart" waypointer. The settings where we use this are suffixed with "S." in the Tables. We observed the planner prefers *closer* goals when obstructed, when the vehicle was already stopped, and when a red light was detected; we observed the planner prefers *farther* goals when unobstructed and when green lights or no lights were observed. Examples of these and other interesting behaviors are best seen in the videos on the website (https://sites.google.com/view/imitative-models). These behaviors follow from the method leveraging $q(\mathbf{S}|\phi)$'s internalization of aspects of expert behavior in order to reproduce them in new situations. Altogether, these results provide affirmative answers to questions (1) and (2). Towards question (3), these results show that our approach is flexible to different directions defined by these goal likelihoods.

Table 3: We evaluate different autonomous driving methods on CARLA's *Dynamic Navigation* task. A "†" indicates methods we have implemented (each observes the same waypoints and LIDAR as input). A "*" indicates results reported in Codevilla et al. (2019). A "–" indicates an unreported statistic. A "‡" indicates an optimistic estimate in transferring a result from the static setting to the dynamic setting. "S." denotes a "smart" waypointer reactive to light state, detailed in Appendix B.2. Results accompanied by standard errors are computed with $N = 3$ trials across environment seeds.

| | Town01 (training conditions) | | | | Town02 (test conditions) | | | |
|---|---|---|---|---|---|---|---|---|
| Dynamic Nav. Method | Success↑ | Ran Red Light↓ | Wrong lane↓ | Off road↓ | Success↑ | Ran Red Light↓ | Wrong lane↓ | Off road↓ |
| CIRL*(Liang et al., 2018) | 82% | – | – | – | 41% | – | – | – |
| CAL*(Sauer et al., 2018) | 83% | – | – | – | 64% | – | – | – |
| MT*(Li et al., 2018) | 81% | – | – | – | 53% | – | – | – |
| CIL*(Codevilla et al., 2018) | 83% | 83%‡ | – | – | 38% | 82%‡ | – | – |
| CILRS*(Codevilla et al., 2019) | 92% | 27%‡ | – | – | 66% | 64%‡ | – | – |
| CILS, Waypoint Input† | 17% | **0.0%** | 0.20% | 12.1% | 36% | **0.0%** | 1.11% | 11.70% |
| MBRL, Waypoint Input† | 64% | 72% | 11.1% | 2.96% | 48% | 54% | 20.6% | 13.3 % |
| *Our method, Region Final-St. Indicator S.*† | 96%±1.9 | **0.89%±0.4** | 0.05%±0.01 | 0.11%±0.01 | 88%±3.3 | 2.60%±0.04 | 0.49%±0.32 | 2.60%±1.1 |
| *Our method, Region Final-St. Indicator*† | 93%±2.2 | 18%±0.5 | 0.023%±0.002 | 0.195%±0.004 | 81%±2.2 | 54.7%±1.5 | 0.12%±0.01 | 1.32%±0.69 |
| *Our method, Line Segment Final-St. Indicator*† | 91%±1.1 | 32%±1.3 | 0.055%±0.002 | 0.013%±0.001 | 88%±3.3 | 35.2%±2.4 | 0.52%±0.03 | 0.18%±0.02 |
| *Our method, Final-State Indicator*† | 92% | 26% | 0.05% | 0.012% | 84% | 35% | 0.13% | 0.38% |
| *Our method, Gaussian Final-St. Mix.*† | 92% | 6.3% | 0.04% | **0.005%** | **100%** | 12% | 0.11% | **0.04%** |
| *Our method, Gaussian Final-St. Mix. S.*† | **100%** | 1.7% | **0.03%** | **0.005%** | 92% | **0.0%** | **0.05%** | 0.15% |

| | Town01 (training conditions) | | | | Town02 (test conditions) | | | |
|---|---|---|---|---|---|---|---|---|
| Static Nav. Method | Success↑ | Ran Red Light↓ | Wrong lane↓ | Off road↓ | Success↑ | Ran Red Light↓ | Wrong lane↓ | Off road↓ |
| CIRL* (Liang et al., 2018) | 93% | – | – | – | 68% | – | – | – |
| CAL* (Sauer et al., 2018) | 92% | – | – | – | 68% | – | – | – |
| MT* (Li et al., 2018) | 81% | – | – | – | 78% | – | – | – |
| CIL* (Codevilla et al., 2018) | 86% | 83% | – | – | 44% | 82% | – | – |
| CILRS*(Codevilla et al., 2019) | 95% | 27% | – | – | 90% | 64% | – | – |
| CILS, Waypoint Input† | 28% | **0.0%** | 0.38% | 10.23% | 36% | **0.0%** | 1.69% | 16.82% |
| MBRL, Waypoint Input† | 96% | 78% | 14.3% | 1.94% | 96% | 73% | 19.6 % | 0.75% |
| *Our method, Final-State Indicator*† | **100%** | 48% | 0.05% | **0.002%** | **100%** | 52% | 0.10% | **0.13%** |
| *Our method, Gaussian Final-St. Mixture*† | 96% | 0.83% | **0.01%** | 0.08% | 96% | **0.0%** | **0.03%** | 0.14% |
| *Our method, Gaussian Final-St. Mix. S.*† | 96% | **0.0%** | 0.04% | 0.07% | 92% | **0.0%** | 0.18% | 0.27% |

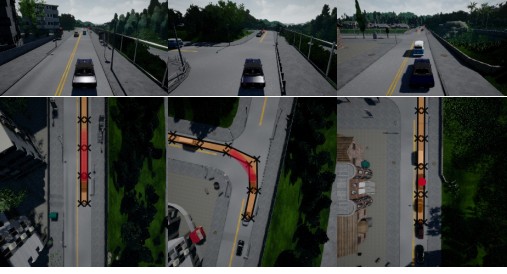

Figure 6: Planning with the Region Final State Indicator yields plans that end inside the region. The orange polygon indicates the region. The red circles indicate the chosen plan.

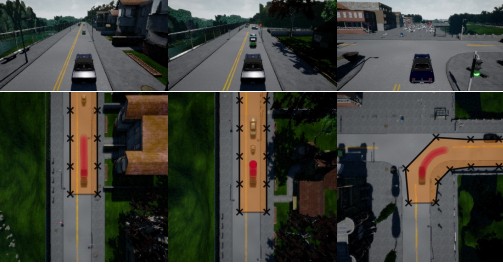

Figure 7: Even with a wider goal region than Fig. 6, the vehicle remains in its lane. Despite their coarseness, these wide goal regions still provide useful guidance to the vehicle.

## 4.1 ROBUSTNESS TO ERRORS IN GOAL-SPECIFICATION

Towards questions (3) (flexibility) and (4) (noise-robustness), we analyze the performance of our method when the path planner is heavily degraded, to understand its stability and reliability. We use the *Gaussian Final-State Mixture* goal likelihood.

**Navigating with high-variance waypoints.** As a test of our model's capability to stay in the distribution of demonstrated behavior, we designed a "decoy waypoints" experiment, in which *half* of the waypoints are highly perturbed versions of the other half, serving as distractions for our Gaussian Final-State Mixture imitative planner. We observed surprising robustness to decoy waypoints. Examples of this robustness are shown in Fig. 10. In Table 4, we report the success rate and the mean number of planning rounds for failed episodes in the "½ distractors" row. These numbers indicate our method can execute dozens of planning rounds without decoy waypoints causing a catastrophic failure, and often it can execute the hundreds necessary to achieve the goal. See Appendix E for details.

**Navigating with waypoints on the wrong side of the road.** We also designed an experiment to test our method under systemic bias in the route planner. Our method is provided waypoints on the

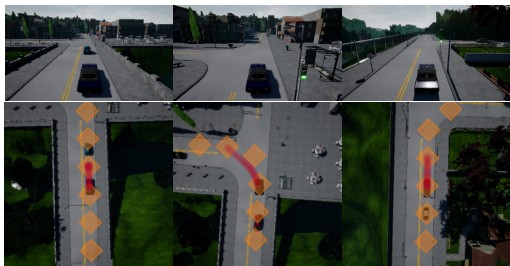 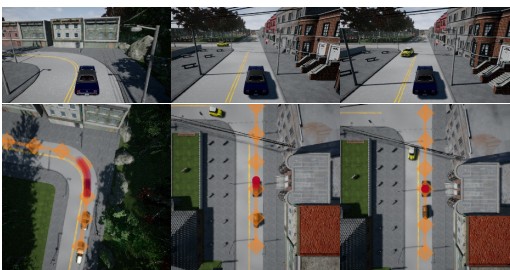

| Figure 8: Planning with the Final State Indicator yields plans that end at one of the provided locations. Orange diamonds indicate the locations in the goal set. Red circles indicate the chosen plan. | Figure 9: Planning with the Line Segment Final State Indicator yields plans that end along a segment. Orange diamonds indicate line segment endpoints. Red circles indicate the chosen plan. |

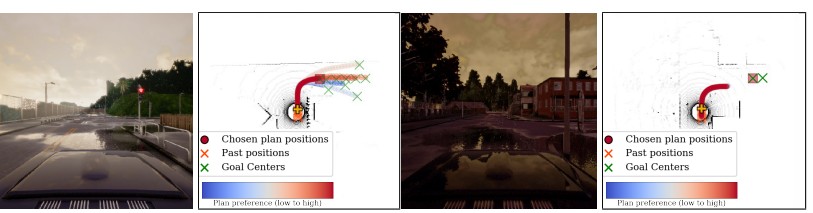 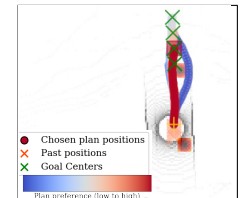

Figure 10: Tolerating bad goals. The planner prefers goals in the distribution of expert behavior (on the road at a reasonable distance). *Left*: Planning with $1/2$ decoy goals. *Right*: Planning with all goals on the wrong side of the road.

Figure 11: Test-time plans steering around potholes.

wrong side of the road (in CARLA, the left side), and tasked with following the directions of these waypoints while staying on the *correct* side of the road (the right side). In order for the value of $q(\mathbf{s}|\phi)$ to outweigh the influence of these waypoints, we increased the $\epsilon$ hyperparameter. We found our method to still be very effective at navigating, and report results in Table 4. We also investigated providing very coarse 8-meter wide regions to the Region Final-State likelihood; these always include space in the wrong lane and off-road (Fig. 7 in Appendix **??** provides visualization). Nonetheless, on Town01 Dynamic, this approach still achieved an overall success rate of $48\%$. Taken together towards question (4), our results indicate that our method is *fairly robust to errors in goal-specification*.

## 4.2 PRODUCING UNOBSERVED BEHAVIORS TO AVOID NOVEL OBSTACLES

Table 4: **Robustness to waypoint noise and test-time pothole adaptation**. Our method is robust to waypoints on the wrong side of the road and fairly robust to decoy waypoints. Our method is flexible enough to safely produce behavior not demonstrated (pothole avoidance) by incorporating a test-time cost. Ten episodes are collected in each Town.

| Waypointer | Extra Cost | Town01 (training conditions) | | | Town02 (test conditions) | | |
|---|---|---|---|---|---|---|---|
| | | Success | Wrong lane | Potholes hit | Success | Wrong lane | Potholes hit |
| Noiseless waypointer | | 100% | 0.00% | 177/230 | 100% | 0.41% | 82/154 |
| Waypoints wrong lane | | 100% | 0.34% | – | 70% | 3.16% | – |
| $1/2$ waypoints distracting | | 70% | – | – | 50% | – | – |
| Noiseless waypointer | Pothole | 90% | 1.53% | 10/230 | 70% | 1.53% | 35/154 |

To further investigate our model's flexibility to test-time objectives (question 3), we designed a pothole avoidance experiment. We simulated potholes in the environment by randomly inserting them in the cost map near waypoints. We ran our method with a test-time-only cost map of the simulated potholes by combining goal likelihoods (F) and (G), and compared to our method that did not incorporate the cost map (using (F) only, and thus had no incentive to avoid potholes). We recorded the number of collisions with potholes. In Table 4, our method with cost incorporated avoided most potholes while avoiding collisions with the environment. To do so, it drove closer to the centerline, and occasionally entered the opposite lane. Our model internalized obstacle avoidance by

staying on the road and demonstrated its flexibility to obstacles not observed during training. Fig. 11 shows an example of this behavior. See Appendix F for details of the pothole generation.

## 5 DISCUSSION

We proposed "Imitative Models" to combine the benefits of IL and MBRL. Imitative Models are probabilistic predictive models able to plan interpretable expert-like trajectories to achieve new goals. Inference with an Imitative Model resembles trajectory optimization in MBRL, enabling it to both *incorporate new goals* and *plan to them* at test-time, which IL cannot. Learning an Imitative Model resembles offline IL, enabling it to circumvent the difficult reward-engineering and costly online data collection necessities of MBRL. We derived families of flexible goal objectives and showed our model can successfully incorporate them without additional training. Our method substantially outperformed six IL approaches and an MBRL approach in a dynamic simulated autonomous driving task. We showed our approach is robust to poorly specified goals, such as goals on the wrong side of the road. We believe our method is broadly applicable in settings where expert demonstrations are available, flexibility to new situations is demanded, and safety is paramount. Future work could investigate methods to handle both observation noise and out-of-distribution observations to enhance the applicability to robust real systems — we expand on this issue in Appendix E. Finally, to facilitate more general planning, future work could extend our approach to explicitly reason about all agents in the environment in order to inform a closed-loop plan for the controlled agent.

## ACKNOWLEDGEMENTS

This research was supported by ONR N000141712623, DARPA Assured Autonomy, ARL DCIST CRA W911NF-17-2-0181, Google, NVIDIA, and Amazon.

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

## A    ALGORITHMS

In Algorithm 1, we provided pseudocode for receding-horizon control via our imitative model. In Algorithm 2 we provided pesudocode that describes how we plan in the latent space of the trajectory. In Algorithm 3, we detail the speed-based throttle and position-based steering PID controllers.

---

**Algorithm 3** PIDCONTROLLER$(\phi = \{\mathbf{s}_0, \mathbf{s}_{-1}, \dots \}, \mathbf{s}_{1:T}^{\mathcal{G}}, h, H; K_p^{\dot{s}}, K_p^{\alpha})$

1: $i \leftarrow T - H + h$ {Compute the index of the target position}
2: $\dot{s}_{\text{process-speed}} \leftarrow (\mathbf{s}_{0,x} - \mathbf{s}_{-1,x})$ {Compute the current forward speed from the observations}
3: $s_{\text{setpoint-position}} \leftarrow \mathbf{s}_{i,x}^{\mathcal{G}}$ {Retrieve the target position x-coordinate from the plan}
4: $\dot{s}_{\text{setpoint-speed}} \leftarrow {}^{s_{\text{setpoint-position}}}/i$ {Compute the forward target speed}
5: $e_{\dot{s}} \leftarrow \dot{s}_{\text{setpoint-speed}} - \dot{s}_{\text{process-speed}}$ {Compute the forward speed error}
6: $u_{\dot{s}} \leftarrow K_p^{\dot{s}} e_{\dot{s}}$ {Compute the accelerator control with a nonzero proportional term}
7: throttle $\leftarrow \mathbb{1}(e > 0) \cdot u + \mathbb{1}(e \le 0) \cdot 0$ {Use the control as throttle if the speed error is positive}
8: brake $\leftarrow \mathbb{1}(e > 0) \cdot 0 + \mathbb{1}(e \le 0) \cdot u$ {Use the control as brake if the speed error is negative}
9: $\alpha_{\text{process}} \leftarrow \arctan(\mathbf{s}_{0,y} - \mathbf{s}_{-1,y}, \mathbf{s}_{0,x} - \mathbf{s}_{-1,x})$ {Compute current heading}
10: $\alpha_{\text{setpoint}} \leftarrow \arctan(\mathbf{s}_{i,y}^{\mathcal{G}} - \mathbf{s}_{0,y}, |\mathbf{s}_{i,x}^{\mathcal{G}} - \mathbf{s}_{0,x}|)$ {Compute target forward heading}
11: $e_{\alpha} \leftarrow \alpha_{\text{setpoint}} - \alpha_{\text{process}}$ {Compute the heading error}
12: steering $\leftarrow K_p^{\alpha} e_{\alpha}$ {Compute the steering with a nonzero proportional term}
13: $u \leftarrow [\text{throttle}, \text{steering}, \text{brake}]$
14: **return** $u$

---

### A.1    LATENT PLAN OPTIMIZATION

Since $\mathbf{s}_{1:T} = f(\mathbf{z}_{1:T})$ in our implementation, and $f$ is differentiable, we can perform gradient descent of the same objective in terms of $\mathbf{z}_{1:T}$, as shown in Algorithm 2.Since $q$ is trained with $\mathbf{z}_{1:T} \sim \mathcal{N}(0, I)$, the latent space is likelier to be better numerically conditioned than the space of $\mathbf{s}_{1:T}$, although we did not compare the two approaches formally. We implemented the following optimizations to improve this procedure's output and practical run time. 1) We started with $N = 120$ different $\mathbf{z}$ initializations, optimized them in batch, and returned the highest-scoring value across the entire optimization. 2) We observed the resulting planning procedure to usually converge quickly, so instead of specifying a convergence threshold, we simply ran the optimization for a small number of steps, $M = 10$, and found that we obtained good performance. Better performance could be obtained by performing a larger number of steps.

## B    GOAL DETAILS

### B.1    OPTIMIZING GOAL LIKELIHOODS WITH SET CONSTRAINTS

We now derive an approach to optimize our main objective with set constraints. Although we could apply a constrained optimizer, we find that we are able to exploit properties of the model and constraints to derive differentiable objectives that enable approximate optimization of the corresponding closed-form optimization problems. These enable us to use the same straightforward gradient-descent-based optimization approach described in Algorithm 2.

**Shorthand notation:**    In this section we omit dependencies on $\phi$ for brevity, and use short hand $\mu_t \doteq \mu_\theta(\mathbf{s}_{1:t-1})$ and $\Sigma_t \doteq \Sigma_\theta(\mathbf{s}_{1:t-1})$. For example, $q(\mathbf{s}_t|\mathbf{s}_{1:t-1}) = \mathcal{N}(\mathbf{s}_t; \mu_t, \Sigma_t)$.

Let us begin by defining a useful delta function:

$$\delta_{\mathbf{s}_T}(\mathbb{G}) \doteq \begin{cases} 1 & \text{if } \mathbf{s}_T \in \mathbb{G} \\ 0 & \text{if } \mathbf{s}_T \notin \mathbb{G}, \end{cases} \tag{4}$$

which serves as our goal likelihood when using goal with set constraints: $p(\mathcal{G}|\mathbf{s}_{1:T}) \leftarrow \delta_{S_T}(\mathbb{G})$. We now derive the corresponding maximum a posteriori optimization problem:

$$
\begin{aligned}
\mathbf{s}_{1:T}^* &\doteq \underset{\mathbf{s}_{1:T}\in\mathbb{R}^{2T}}{\arg\max}\ p(\mathbf{s}_{1:T}|\mathcal{G}) \\
&= \underset{\mathbf{s}_{1:T}\in\mathbb{R}^{2T}}{\arg\max}\ p(\mathcal{G}|\mathbf{s}_{1:T}) \cdot q(\mathbf{s}_{1:T}) \cdot p^{-1}(\mathcal{G}) \\
&= \underset{\mathbf{s}_{1:T}\in\mathbb{R}^{2T}}{\arg\max}\ \underbrace{p(\mathcal{G}|\mathbf{s}_{1:T})}_{\text{goal likelihood}} \cdot \underbrace{q(\mathbf{s}_{1:T})}_{\text{imitation prior}} \\
&= \underset{\mathbf{s}_{1:T}\in\mathbb{R}^{2T}}{\arg\max}\ \underbrace{\delta_{S_T}(\mathbb{G})}_{\text{set constraint}} \cdot \underbrace{q(\mathbf{s}_{1:T})}_{\text{imitation prior}} \\
&= \underset{\mathbf{s}_{1:T}\in\mathbb{R}^{2T}}{\arg\max}\ \begin{cases} q(\mathbf{s}_{1:T}) & \text{if } \mathbf{s}_T \in \mathbb{G} \\ 0 & \text{if } \mathbf{s}_T \notin \mathbb{G} \end{cases} \\
&= \underset{\mathbf{s}_{1:T-1}\in\mathbb{R}^{2(T-1)}, \mathbf{s}_T\in\mathbb{G}}{\arg\max}\ q(\mathbf{s}_{1:T}) \\
&= \underset{\mathbf{s}_{1:T-1}\in\mathbb{R}^{2(T-1)}}{\arg\max}\ \underset{\mathbf{s}_T\in\mathcal{G}}{\arg\max}\ q(\mathbf{s}_T|\mathbf{s}_{1:T-1})\prod_{t=1}^{T-1}q(\mathbf{s}_t|\mathbf{s}_{1:t-1}) \\
&= \underset{\mathbf{s}_{1:T-1}\in\mathbb{R}^{2(T-1)}}{\arg\max}\ \underset{\mathbf{s}_T\in\mathcal{G}}{\arg\max}\ \mathcal{N}(\mathbf{s}_T;\mu_T,\Sigma_T)\prod_{t=1}^{T-1}\mathcal{N}(\mathbf{s}_t;\mu_t,\Sigma_t).
\end{aligned}
\tag{5}
$$

By exploiting the fact that $q(\mathbf{s}_T|\mathbf{s}_{1:T-1}) = \mathcal{N}(\mathbf{s}_T;\mu_T,\Sigma_T)$, we can derive closed-form solutions for

$$
\mathbf{s}_T^* = \underset{\mathbf{s}_T\in\mathbb{G}}{\arg\max}\ \mathcal{N}(\mathbf{s}_T;\mu_T,\Sigma_T)
\tag{6}
$$

when $\mathbb{G}$ has special structure, which *enables us to apply gradient descent to solve this constrained-optimization problem* (examples below). With a closed form solution to equation 6, we can easily compute equation 5 using *unconstrained*-optimization as follows:

$$
\mathbf{s}_{1:T}^* = \underset{\mathbf{s}_{1:T-1}\in\mathbb{R}^{2(T-1)}}{\arg\max}\ \underset{\mathbf{s}_T\in\mathbb{G}_{\text{line-segment}}}{\arg\max}\ q(\mathbf{s}_T|\mathbf{s}_{1:T-1})\prod_{t=1}^{T-1}q(\mathbf{s}_t|\mathbf{s}_{1:t-1})
\tag{7}
$$

$$
\mathbf{s}_{1:T-1}^* = \underbrace{\underset{\mathbf{s}_{1:T-1}\in\mathbb{R}^{2(T-1)}}{\arg\max}}_{\text{unconstrained optimization}}\ \underbrace{q(\mathbf{s}_T^*|\mathbf{s}_{1:t-1})\prod_{t=1}^{T-1}q(\mathbf{s}_t|\mathbf{s}_{1:t-1})}_{\text{objective function of } \mathbf{s}_{1:T-1}}.
\tag{8}
$$

Note that equation 8 only helps solve equation 5 if equation 6 has a closed-form solution. We detail example of goal-sets with such closed-form solutions in the following subsections.

### B.1.1 POINT GOAL-SET

The solution to equation 6 in the case of a single desired goal $g \in \mathbb{R}^D$ is simply:

$$
\mathbb{G}_{\text{point}} \doteq \{\mathbf{g}_T\},
\tag{9}
$$

$$
\mathbf{s}_{T,\text{point}}^* \doteq \underset{\mathbf{s}_T\in\mathbb{G}_{\text{point}}}{\arg\max}\ \mathcal{N}(\mathbf{s}_T;\mu_T,\Sigma_T)
$$

$$
= \mathbf{g}_T.
\tag{10}
$$

More generally, multiple point goals help define *optional* end points for planning: where the agent only need reach one valid end point (see Fig. 8 for examples), formulated as:

$$
\mathbb{G}_{\text{points}} \doteq \{\mathbf{g}_T^k\}_{k=1}^K,
\tag{11}
$$

$$
\mathbf{s}_{T,\text{points}}^* \doteq \underset{\mathbf{g}_T^k\in\mathbb{G}_{\text{points}}}{\arg\max}\ \mathcal{N}\left(\mathbf{g}_T^k;\mu_T,\Sigma_T\right).
\tag{12}
$$

### B.1.2 Line-segment goal-set

We can form a goal-set as a finite-length line segment, connecting point $\mathbf{a} \in \mathbb{R}^D$ to point $\mathbf{b} \in \mathbb{R}^D$:

$$g_{\text{line}}(u) \doteq \mathbf{a} + u \cdot (\mathbf{b} - \mathbf{a}), \ u \in \mathbb{R}, \tag{13}$$

$$\mathbb{G}^{\mathbf{a} \rightarrow \mathbf{b}}_{\text{line-segment}} \doteq \{g_{\text{line}}(u) : u \in [0, 1]\}. \tag{14}$$

The solution to equation 6 in the case of line-segment goals is:

$$\mathbf{s}^*_{T, \text{line-segment}} \doteq \underset{\mathbf{s}_T \in \mathbb{G}^{\mathbf{a} \rightarrow \mathbf{b}}_{\text{line-segment}}}{\arg\max} \ \mathcal{N}(\mathbf{s}_T; \mu_T, \Sigma_T) \tag{15}$$

$$= \mathbf{a} + \min\left(1, \ \max\left(0, \ \frac{(\mathbf{b} - \mathbf{a})^\top \Sigma_T^{-1}(\mu_T - \mathbf{a})}{(\mathbf{b} - \mathbf{a})^\top \Sigma_T^{-1}(\mathbf{b} - \mathbf{a})}\right)\right) \cdot (\mathbf{b} - \mathbf{a}). \tag{16}$$

**Proof:**

To solve equation 15 is to find which point along the line $g_{\text{line}}(u)$ maximizes $\mathcal{N}(\cdot; \mu_T, \Sigma_T)$ subject to the constraint $0 \leq u \leq 1$:

$$u^* \doteq \underset{u \in [0,1]}{\arg\max} \ \mathcal{N}(g_{\text{line}}(u); \mu_T, \Sigma_T))$$

$$= \underset{u \in [0,1]}{\arg\min} \ \underbrace{(g_{\text{line}}(u) - \mu_T)^\top \Sigma_T^{-1}(g_{\text{line}}(u) - \mu_T)}_{\mathcal{L}_u(u)}. \tag{17}$$

Since $\mathcal{L}_u$ is convex, the optimal value $u^*$ is value closest to the unconstrained $\arg\max$ of $\mathcal{L}_u(u)$, subject to $0 \leq u \leq 1$:

$$u^*_{\mathbb{R}} \doteq \underset{u \in \mathbb{R}}{\arg\max} \ \mathcal{L}_u(u), \tag{18}$$

$$u^* = \underset{u \in [0,1]}{\arg\min} \ \mathcal{L}_u(u)$$

$$= \min(1, \ \max(0, \ u^*_{\mathbb{R}})). \tag{19}$$

We now solve for $u^*_{\mathbb{R}}$:

$$u^*_{\mathbb{R}} = u : 0 = \frac{d\mathcal{L}(u)}{du} = \frac{d\left((g_{\text{line}}(u) - \mu_T)^\top \Sigma_T^{-1}(g_{\text{line}}(u) - \mu_T)\right)}{du}$$

$$= 2 \cdot \frac{d(g_{\text{line}}(u) - \mu_T)^\top}{du} \Sigma_T^{-1}(g_{\text{line}}(u) - \mu_T)$$

$$= 2 \cdot \frac{d(\mathbf{a} + u \cdot (\mathbf{b} - \mathbf{a}) - \mu_T)^\top}{du} \Sigma_T^{-1}(\mathbf{a} + u \cdot (\mathbf{b} - \mathbf{a}) - \mu_T)$$

$$= 2 \cdot (\mathbf{b} - \mathbf{a})^\top \Sigma_T^{-1}(\mathbf{a} + u \cdot (\mathbf{b} - \mathbf{a}) - \mu_T),$$

$$u^*_{\mathbb{R}} = \frac{(\mathbf{b} - \mathbf{a})^\top \Sigma_T^{-1}(\mu_T - \mathbf{a})}{(\mathbf{b} - \mathbf{a})^\top \Sigma_T^{-1}(\mathbf{b} - \mathbf{a})}, \tag{20}$$

which gives us:

$$\mathbf{s}^*_{T, \text{line-segment}} = g_{\text{line}}(u^*)$$

$$= \mathbf{a} + u^* \cdot (\mathbf{b} - \mathbf{a})$$

$$= \mathbf{a} + \min(1, \ \max(0, \ u^*_{\mathbb{R}})) \cdot (\mathbf{b} - \mathbf{a})$$

$$= \mathbf{a} + \min\left(1, \ \max\left(0, \ \frac{(\mathbf{b} - \mathbf{a})^\top \Sigma_T^{-1}(\mu_T - \mathbf{a})}{(\mathbf{b} - \mathbf{a})^\top \Sigma_T^{-1}(\mathbf{b} - \mathbf{a})}\right)\right) \cdot (\mathbf{b} - \mathbf{a}). \tag{21}$$

### B.1.3 Multiple-line-segment goal-set:

More generally, we can combine multiple line-segments to form piecewise linear "paths" we wish to follow. By defining a path that connects points $(\mathbf{p}_0, \mathbf{p}_1, ..., \mathbf{p}_N)$, we can evaluate $\mathcal{L}^i_u$ for each $\mathbb{G}^{\mathbf{p}_i \rightarrow \mathbf{p}_{i+1}}_{\text{line-segment}}$, select the optimal segment $i^* = \arg\max_i \mathcal{L}^i_u$, and use the segment $i^*$'s solution to $u^*$ to compute $s^*_T$. Examples shown in Fig. 9.

### B.1.4 Polygon goal-set

Instead of a route or path, a user (or program) may wish to provide a general *region* the agent should go to, and state within that region being equally valid. Polygon regions (including both boundary and interior) offer closed form solution to equation 6 and are simple to specify. A polygon can be specified by an ordered sequence of vertices $(\mathbf{p}_0, \mathbf{p}_1, ..., \mathbf{p}_N) \in \mathbb{R}^{N \times 2}$. Edges are then defined as the sequence of line-segments between successive vertices (and a final edge between first and last vertex): $((\mathbf{p}_0, \mathbf{p}_1), ..., (\mathbf{p}_{N-1}, \mathbf{p}_N), (\mathbf{p}_N, \mathbf{p}_0))$. Examples shown in Fig. 6 and 7.

Solving equation 6 with a polygon has two cases: depending whether $\mu_T$ is *inside* the polygon, or *outside*. If $\mu_T$ lies inside the polygon, then the optimal value for $\mathbf{s}_T^*$ that maximizes $\mathcal{N}(\mathbf{s}_T^*; \mu_T, \Sigma_T)$ is simply $\mu_T$: the mode of the Gaussian distribution. Otherwise, if $\mu_T$ lies outside the polygon, then the optimal value $\mathbf{s}_T^*$ will lie on one of the polygon's edges, solved using B.1.3.

### B.2 Waypointer Details

The waypointer uses the CARLA planner's provided route to generate waypoints. In the constrained-based planning goal likelihoods, we use this route to generate waypoints without interpolating between them. In the relaxed goal likelihoods, we interpolate this route to every 2 meters, and use the first 20 waypoints. As mentioned in the main text, one variant of our approach uses a "smart" waypointer. This waypointer simply removes nearby waypoints closer than 5 meters from the vehicle when a green light is observed in the measurements provided by CARLA, to encourage the agent to move forward, and removes far waypoints beyond 5 meters from the vehicle when a red light is observed in the measurements provided by CARLA. Note that the performance differences between our method without the smart waypointer and our method with the smart waypointer are small: the only signal in the metrics is that the smart waypointer improves the vehicle's ability to stop for red lights, however, it is quite adept at doing so without the smart waypointer.

### B.3 Constructing Goal Sets

Given the in-lane waypoints generated by CARLA's route planner, we use these to create Point goal sets, Line-Segment goal sets, and Polygon Goal-Sets, which respectively correspond to the (A) Final-State Indicator, (B) Line-Segment Final-State Indicator, and (C) Final-State Region Indicator described in Section 2.2. For (A), we simply feed the waypoints directly into the Final-State Indicator, which results in a constrained optimization to ensure that $S_T \in \mathbb{G} = \{g_T^k\}_{k=1}^K$. We also included the vehicle's current position in the goal set, in order to allow it to stop. The gradient-descent based optimization is then formed from combining Eq. 8 with Eq. 12. The gradient to the nearest goal of the final state of the partially-optimized plan encourage the optimization to move the plan closer to that goal. We used $K = 10$. We applied the same procedure to generate the goal set for the (B) Line Segment indicator, as the waypoints returned by the planner are ordered. Finally, for the (C) Final-State Region Indicator (polygon), we used the ordered waypoints as the "skeleton" of a polygon that surrounds. It was created by adding a two vertices for each point $\mathbf{v}_t$ in the skeleton at a distance 1 meter from $\mathbf{v}_t$ perpendicular to the segment connecting the surrounding points $(\mathbf{v}_{t-1}, \mathbf{v}_{t+1})$. This resulted in a goal set $\mathbb{G}_{\text{polygon}} \supset \mathbb{G}_{\text{line-segment}}$, as it surrounds the line segments. The (F) Gaussian Final-State Mixture goal set was constructed in the same way as (A), and also used when the pothole costs were added.

For the methods we implemented, the task is to drive the *furthest* road location from the vehicle's initial position. Note that this protocol more difficult than the one used in prior work Codevilla et al. (2018); Liang et al. (2018); Sauer et al. (2018); Li et al. (2018); Codevilla et al. (2019), which has no distance guarantees between start positions and goals, and often results in shorter paths.

## C Architecture and Training Details

The architecture of $q(\mathbf{S}|\phi)$ is shown in Table 5.

### C.1 Prior Visualization and Statistics

We show examples of the priors multimodality in Fig. 13

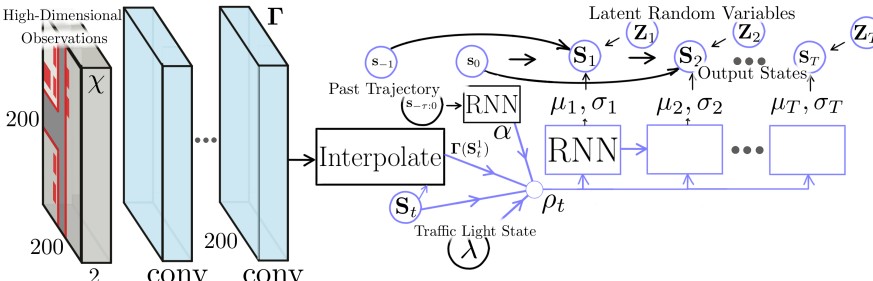

Figure 12: Architecture of $m_\theta$ and $\sigma_\theta$, which parameterize $q_\theta(\mathbf{S}|\phi = \{\chi, \mathbf{s}_{-\tau:0}, \boldsymbol{\lambda}\})$. Inputs: LIDAR $\chi$, past-states $\mathbf{s}_{-\tau:0}$, light-state $\boldsymbol{\lambda}$, and latent noise $\mathbf{Z}_{1:T}$. Output: trajectory $\mathbf{S}_{1:T}$. Details in Appendix C.

Table 5: Detailed Architecture that implements $\mathbf{s}_{1:T} = f(\mathbf{z}_{1:T}, \phi)$. Typically, $T = 40$, $D = 2, H = W = 200$.

| Component | Input [dimensionality] | Layer or Operation | Output [dimensionality] | Details |
|---|---|---|---|---|
| *Static featurization of context:* $\phi = \{\chi, \mathbf{s}_{-\tau:0}^{1:A}\}$. | | | | |
| MapFeat | $\chi\,[H, W, 2]$ | 2D Convolution | $^1\chi\,[H, W, 32]$ | $3 \times 3$ stride 1, ReLu |
| MapFeat | $^{i-1}\chi\,[H, W, 32]$ | 2D Convolution | $^i\chi\,[H, W, 32]$ | $3 \times 3$ stride 1, ReLu, $i \in [2, \ldots, 8]$ |
| MapFeat | $^8\chi\,[H, W, 32]$ | 2D Convolution | $\boldsymbol{\Gamma}\,[H, W, 8]$ | $3 \times 3$ stride 1, ReLu |
| PastRNN | $\mathbf{s}_{-\tau:0}\,[\tau + 1, D]$ | RNN | $\alpha\,[32]$ | GRU across time dimension |
| *Dynamic generation via loop:* for $t \in \{0, \ldots, T-1\}$. | | | | |
| MapFeat | $\mathbf{s}_t\,[D]$ | Interpolate | $\gamma_t = \boldsymbol{\Gamma}(\mathbf{s}_t)\,[8]$ | Differentiable interpolation |
| JointFeat | $\gamma_t, \mathbf{s}_0, {}^2\eta, \alpha, \boldsymbol{\lambda}$ | $\gamma_t \oplus \mathbf{s}_0 \oplus {}^2\eta \oplus \alpha \oplus \boldsymbol{\lambda}$ | $\rho_t\,[D + 50 + 32 + 1]$ | Concatenate ($\oplus$) |
| FutureRNN | $\rho_t\,[D + 50 + 32 + 1]$ | RNN | $^1\rho_t\,[50]$ | GRU |
| FutureMLP | $^1\rho_t\,[50]$ | Affine (FC) | $^2\rho_t\,[200]$ | Tanh activation |
| FutureMLP | $^2\rho_t\,[200]$ | Affine (FC) | $m_t\,[D],\ \xi_t\,[D, D]$ | Identity activation |
| MatrixExp | $\xi_t\,[D, D]$ | $\mathrm{expm}(\xi_t + \xi_t^{a, \text{transpose}})$ | $\boldsymbol{\sigma}_t\,[D, D]$ | Differentiable Matrix Exponential Rhinehart et al. (2018) |
| VerletStep | $\mathbf{s}_t, \mathbf{s}_{t-1}, m_t, \boldsymbol{\sigma}_t, \mathbf{z}_t$ | $2\mathbf{s}_t - \mathbf{s}_{t-1} + m_t + \boldsymbol{\sigma}_t \mathbf{z}_t$ | $\mathbf{s}_{t+1}\,[D]$ | |

### C.1.1 STATISTICS OF PRIOR AND GOAL LIKELIHOODS

Following are the values of the planning criterion on $N \approx 8 \cdot 10^3$ rounds from applying the "Gaussian Final-State Mixture" to Town01 Dynamic. Mean of $\log q(\mathbf{s}^*|\phi) \approx 104$. Mean of $\log p(\mathcal{G}|\mathbf{s}^*, \phi) = -4$. This illustrates that while the prior's value mostly dominates the values of the final plans, the Gaussian Final-State Goal Mixture likelihood has a moderate amount of influence on the value of the final plan.

### C.2 DATASET

Before training $q(\mathbf{S}|\phi)$, we ran CARLA's expert in the dynamic world setting of Town01 to collect a dataset of examples. We have prepared the dataset of collected data for public release upon publication. We ran the autopilot in Town01 for over 900 episodes of 100 seconds each in the presence of 100 other vehicles, and recorded the trajectory of every vehicle and the autopilot's LIDAR observation. We randomized episodes to either train, validation, or test sets. We created sets of 60,701 train, 7586 validation, and 7567 test scenes, each with 2 seconds of past and 4 seconds of future position information at 10Hz. The dataset also includes 100 episodes obtained by following the same procedure in Town02.

## D BASELINE DETAILS

### D.1 CONDITIONAL IMITATION LEARNING OF STATES (CILS):

We designed a conditional imitation learning baseline that predicts the setpoint for the PID-controller. Each receives the same scene observations (LIDAR) and is trained with the same set of trajectories as our main method. It uses nearly the same architecture as that of the original CIL, except it outputs setpoints instead of controls, and also observes the traffic light information. We found it very effective for stable control on straightaways. When the model encounters corners, however, prediction is more

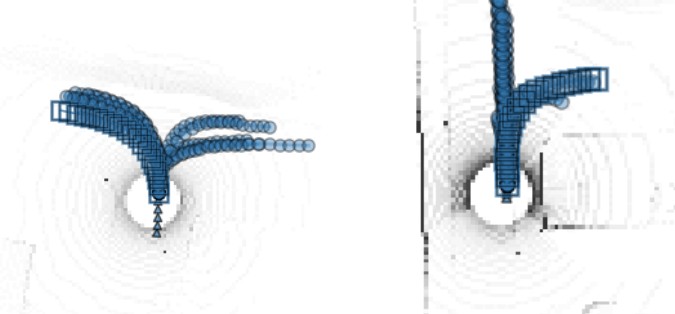

Figure 13: *Left*: Samples from the prior, $q(\mathbf{S}|\phi)$, go left or right. *Right:* Samples go forward or right.

difficult, as in order to successfully avoid the curbs, the model must implicitly plan a safe path. We found that using the traffic light information allowed it to stop more frequently.

### D.2 MODEL-BASED REINFORCEMENT LEARNING:

**Static-world** To compare against a purely model-based reinforcement learning algorithm, we propose a model-based reinforcement learning baseline. This baseline first learns a forwards dynamics model $\mathbf{s}_{t+1} = f(\mathbf{s}_{t-3:t}, \mathbf{a}_t)$ given observed expert data ($a_t$ are recorded vehicle actions). We use an MLP with two hidden layers, each 100 units. Note that our forwards dynamics model does not imitate the expert preferred actions, but only models what is physically possible. Together with the same LIDAR map $\chi$ our method uses to locate obstacles, this baseline uses its dynamics model to plan a reachability tree LaValle (2006) through the free-space to the waypoint while avoiding obstacles. The planner opts for the lowest-cost path that ends near the goal $C(\mathbf{s}_{1:T}; \mathbf{g}_T) = ||\mathbf{s}_T - \mathbf{g}_T||_2 + \sum_{t=1}^{T} c(\mathbf{s}_t)$, where cost of a position is determined by $c(\mathbf{s}_t) = 1.5\mathbb{1}(\mathbf{s}_t < 1 \text{ meters from any obstacle}) + 0.75\mathbb{1}(1 <= \mathbf{s}_t < 2 \text{ meters from any obstacle}) + \ddot{\mathbf{s}}_t$.

We plan forwards over 20 time steps using a breadth-first search over CARLA steering angle $\{-0.3, -0.1, 0., 0.1, 0.3\}$, noting valid steering angles are normalized to $[-1, 1]$, with constant throttle at 0.5, noting the valid throttle range is $[0, 1]$. Our search expands each state node by the available actions and retains the 50 closest nodes to the waypoint. The planned trajectory efficiently reaches the waypoint, and can successfully plan around perceived obstacles to avoid getting stuck. To convert the LIDAR images into obstacle maps, we expanded all obstacles by the approximate radius of the car, 1.5 meters.

**Dynamic-world** We use the same setup as the Static-MBRL method, except we add a discrete temporal dimension to the search space (one $\mathbb{R}^2$ spatial dimension per T time steps into the future). All static obstacles remain static, however all LIDAR points that were known to collide with a vehicle are now removed: and replaced at every time step using a constant velocity model of that vehicle. We found that the main failure mode was due to both to inaccuracy in constant velocity prediction as well as the model's inability to perceive lanes in the LIDAR. The vehicle would sometimes wander into the opposing traffic's lane, having failed to anticipate an oncoming vehicle blocking its path.

## E ROBUSTNESS

### E.1 DECOY WAYPOINTS EXPERIMENTS

In the decoy waypoints experiment, the perturbation distribution is $\mathcal{N}(0, \sigma = 8m)$: each waypoint is perturbed with a standard deviation of 8 meters. One failure mode of this approach is when decoy waypoints lie on a valid off-route path at intersections, which temporarily confuses the planner about the best route. Additional visualizations are shown in Fig. 14.

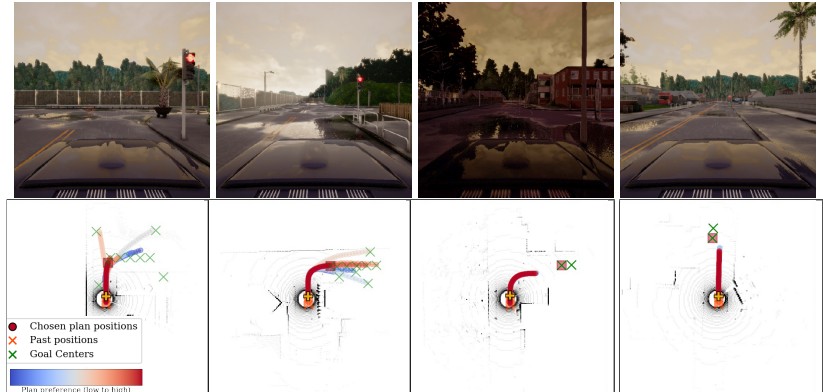

Figure 14: Tolerating bad waypoints. The planner prefers waypoints in the distribution of expert behavior (on the road at a reasonable distance). *Columns 1,2*: Planning with 1/2 decoy waypoints. *Columns 3,4*: Planning with all waypoints on the wrong side of the road.

### E.2    PLAN RELIABILITY ESTIMATION

Besides using our model to make a best-effort attempt to reach a user-specified goal, the fact that our model produces explicit likelihoods can also be leveraged to test the *reliability* of a plan by evaluating whether reaching particular waypoints will result in human-like behavior or not. This capability can be quite important for real-world safety-critical applications, such as autonomous driving, and can be used to build a degree of fault tolerance into the system. We designed a classification experiment to evaluate how well our model can recognize safe and unsafe plans. We planned our model to known good waypoints (where the expert actually went) and known bad waypoints (off-road) on 1650 held-out test scenes. We used the planning criterion to classify these as good and bad plans and found that we can detect these bad plans with $97.5\%$ recall and $90.2\%$ precision. This result indicates imitative models could be effective in estimating the reliability of plans.

We determined a threshold on the planning criterion by single-goal planning to the expert's final location on offline validation data and setting it to the criterion's mean minus one stddev. Although a more intelligent calibration could be performed by analyzing the information retrieval statistics on the offline validation, we found this simple calibration to yield reasonably good performance. We used 1650 test scenes to perform classification of plans to three different types of waypoints 1) where the expert actually arrived at time $T$ ($89.4\%$ reliable), 2) waypoints 20m ahead along the waypointer-provided route, which are often near where the expert arrives ($73.8\%$ reliable) 3) the same waypoints from 2), shifted 2.5m off of the road ($2.5\%$ reliable). This shows that our learned model exhibits a strong preference for valid waypoints. Therefore, a waypointer that provides expert waypoints via 1) half of the time, and slightly out-of-distribution waypoints via 3) in the other half, an "unreliable" plan classifier achieves $97.5\%$ recall and $90.2\%$ precision.

### E.3    OUT-OF-DISTRIBUTION ROBUSTNESS

The existence of both (1) observation noise and (2) uncertain/out-of-distribution observations is an important practical issue for autonomous vehicles. Although our current method only conditions on our current observation, several extensions could help mitigate the negative effects of both (1) and (2). For (1), a Bayesian filtering formulation is arguably most ideal, to better estimate (and track) the location of static and dynamic obstacles under noise. However, such high-dimensional filtering are often intractable, and might necessitate approximate Bayesian deep learning techniques, RNNs, or frame stacking, to benefit from multiple observations. Addressing (2) would ideally be done by placing a prior over our neural network weights, to derive some measure of confidence in our density estimation of how expert each plan is, such that unfamiliar scenes generate large uncertainty on our density estimate that we could detect, and react cautiously (pessimistically) to. One way to address the situation if the distributions are very different is to adopt an ensembling approach Lakshminarayanan et al. (2017) in order for the method to determine when the inputs are out of distribution — the ensemble will usually have higher variance (i.e. disagree) when each element of

the ensemble is provided with an out-of-distribution input. For instance, this variance could be used as a penalization in the planning criterion.

### E.4 Traffic-Light Noise

As discussed, our model assumes access to the traffic-light state provided by the simulator, which we call $\lambda$. However, access to this state would be noisy in practice, because it relies on a sensor-based (usually image-based) detection and classification module.

We performed an experiment to assess robustness to noise in $\lambda$: we simulated noise in $\lambda$ by "flipping" the light state with 20% probability, corresponding to a light state detector that has 80% accuracy on average. "Flipping" means that if the light is green, then changing $\lambda$ to indicate red, and if the light is red, then changing $\lambda$ to indicate green. We performed this following the experimental method of "Region Final-St. Indicator S." in dynamic Town02, and ran it with three separate seeds. The means and their standard errors are reported in Table 6. The conclusion we draw is that the approach can still achieve success most of the time, although it tends to violate red-lights more often. Qualitatively, we observed the resulting behavior near intersections to sometimes be "jerky", with the model alternating between stopping and non-stopping plans. We hypothesize that the model itself could be made more robust if the noise in $\lambda$ was also present in the training data.

Table 6: We evaluate the effect of noise in the traffic-light state ($\lambda$) on CARLA's *Dynamic Navigation* task. Noise in the light state predictably degrades overall and red-light performance, but not to the point of preventing the method from operating at all.

| Town02 Dynamic Navigation Method | Success | Ran Red Light | Wrong lane | Off road |
|---|---|---|---|---|
| *Region Final-St. Indicator S.* (original) | $88\% \pm 3.3$ | $2.57\% \pm 0.04$ | $0.49\% \pm 0.32$ | $2.6\% \pm 1.06$ |
| *Region Final-St. Indicator S.* (noisy $\lambda$) | $76\% \pm 5.0$ | $34.8\% \pm 2.4$ | $0.15\% \pm 0.04$ | $1.79\% \pm 0.34$ |

## F  Pothole Experiment Details

We simulated potholes in the environment by randomly inserting them in the cost map near each waypoint $i$ with offsets distributed $\mathcal{N}_i(\mu=[-15\text{m}, 0\text{m}], \Sigma = \text{diag}([1, 0.01]))$, (i.e. mean-centered on the right side of the lane $15$m before each waypoint). We inserted pixels of root cost $-1e3$ in the cost map at a single sample of each $\mathcal{N}_i$, binary-dilated the cost map by $1/3$ of the lane-width (spreading the cost to neighboring pixels), and then blurred the cost map by convolving with a normalized truncated Gaussian filter of $\sigma = 1$ and truncation width 1.

## G  Baseline Visualizations

See Fig. 15 for a visualization of our baseline methods.

## H  Hyperparameters

In order to tune the $\epsilon$ hyperparameter of the unconstrained likelihoods, we undertook the following binary-search procedure. When the prior frequently overwhelmed the posterior, we set $\epsilon \leftarrow 0.2\epsilon$, to yield tighter covariances, and thus *more* penalty for failing to satisfy the goals. When the posterior frequently overwhelmed the prior, we set $\epsilon \leftarrow 5\epsilon$, to yield looser covariances, and thus *less* penalty for failing to satisfy the goals. We executed this process three times: once for the "Gaussian Final-State Mixture" experiments (Section 4), once for the "Noise Robustness" Experiments (Section 4.1), and once for the pothole-planning experiments (Section 4.2). Note that the Constrained-Goal Likelihoods introduced no hyperparameters to tune.

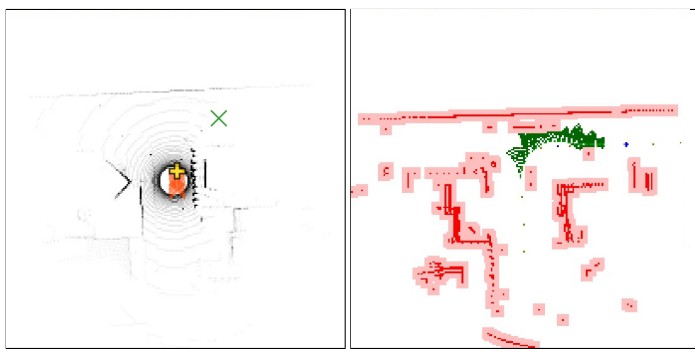

Figure 15: Baseline methods we compare against. The red crosses indicate the past 10 positions of the agent. *Left: Imitation Learning baseline*: the green cross indicates the provided goal, and the yellow plus indicates the predicted setpoint for the controller. *Right: Model-based RL baseline:* the green regions indicate the model's predicted reachability, the red regions are post-processed LIDAR used to create its obstacle map.

