# OpenReview forum: "Deep Imitative Models for Flexible Inference, Planning, and Control"
_ICLR.cc/2020/Conference — Accept (Poster)_

### Official Review · AnonReviewer3 · 2019-10-20
**Official Blind Review #3**

**Rating:** 8

**Review:**

Summary:
- key problem: expert-like probabilistic online motion planning to reach arbitrary goals without reward shaping thanks to off-line learning from expert demonstrations;
- contributions: 1) an imitative planning procedure via gradient-based log-likelihood maximization leveraging "imitative models" q(future states | features), 2) multiple proposals to define flexible goals in this probabilistic framework, 3) a complete implementation for end-to-end navigation in CARLA, 4) an extensive experimental evaluation showcasing the performance, flexibility, interpretability, and robustness of the proposed approach w.r.t. the previous state of the art and several Imitation Learning (IL) and Model-Based Reinforcement Learning (MBRL) baselines.

Recommendation: weak accept (leaning towards strong accept)

Key reason 1: principled probabilistic framework bringing the best of both IL and MBRL worlds.
- this planning as inference method is very succinctly and elegantly described in the paper with enough details in appendix (+ code) to suggest a high chance of reproducibility;
- the flexibility of defining different interpretable goals (6 different types explored in the paper) highlights the versatility of the approach;
- the additional benefits in terms of plan reliability estimation (Appendix E) are significant;
- the paper showcases how powerful and useful a good "imitative model" can be, therefore, reinforcing the interest of the research community in the important topic of off-line learning from large datasets of demonstrations (without requiring costly on-line data collection).

Key reason 2: thorough experimental evaluation with convincing results.
- the experimental protocol used is the standard one on CARLA and the results are state of the art;
- the comparison with related works, including recent ones, is thorough and well explained;
- the additional claims regarding robustness are substantiated by multiple experiments.

Suggested improvements:
- Not needing reward engineering is a major claim of this approach, but it seems that constructing goal likelihoods could be seen as a form of reward engineering, no? Table 3 indeed reports significant performance differences (absolute and relative) depending on how the goals are specified, especially in dynamic environments. As the experiments aim at maximizing the same performance metrics, is there a preferred goal type that works well across all experiments? If not, how is "goal definition" different than "reward engineering"?
- Could the authors please include a variance analysis (using different seeds) in Tables 3 and 4? Previous papers have reported high variance in similar settings (cf., Codevilla et al 2019), and this is a common issue in IL/RL.
- How important is knowing \lambda (traffic light state) perfectly in practice? Can the robustness to noise in \lambda be experimentally assessed? I would also clarify in section 4 and Table 2 that other methods do not use \lambda (the traffic light state), which is a signal very strongly correlated with the "ran red light" metric.
- More generally, what is the robustness of this approach to uncertainty / noise in \phi? Although it is typically available (as the authors mentioned) it is never perfect in practice. Can this be handled in a principled probabilistic way as an extension of the current formulation?
- The current model does not factor the influence of the agent on its environment (\phi := \phi_{t=0}). Is this framework limited to open loop planning, or does this open interesting future research directions towards closing the loop? It seems to be a key open problem to at least discuss in Section 5.

Additional Feedback:
- Figure 5 is confusing, not sure it adds much value to the paper;
- typos in Appendix ("pesudocode", "baselines that predicts", "search search").

 ## Update post rebuttal

Thanks to the authors' excellent replies and my initial inclination towards strong accept, I am happy to bump my score to 8. The authors did an excellent job, their rebuttal is on point, not avoiding hard questions, running additional requested experiments (incl. in a clever way for the most computationally expensive ones), and showing clear insights in future steps. Great job!

**Experience Assessment:**

I have published one or two papers in this area.

**Review Assessment: Checking Correctness Of Derivations And Theory:**

I assessed the sensibility of the derivations and theory.

**Review Assessment: Checking Correctness Of Experiments:**

I carefully checked the experiments.

**Review Assessment: Thoroughness In Paper Reading:**

I read the paper thoroughly.

---

> ### Author Response · Authors · 2019-11-12
> **Response to Review 3 (part 1)**
>
> We thank R3 for their insightful and detailed review, constructive feedback for clarity and additional experiments, and favorable impression of the method.
>
> > “Constructing goal likelihoods could be seen as a form of reward engineering...Is there a preferred goal type that works well across all experiments?”
>
> Yes, you are right that designing general goal likelihoods can be considered a form of reward engineering if there are no restrictions on the goal likelihoods. We have incorporated elements of the following discussion into Section 2.2 in order to make this point clearer. The connection between goal likelihoods and reward functions is best seen in (G) the energy-based likelihood, which has an explicit cost term. One reason why it is easier to design goal likelihoods is that the task of evoking most aspects of goal-driven behavior is already learned by the prior q(s|\phi), which models desirable behavior. This is in contrast to model-free reinforcement learning, which entirely relies on the reward design to evoke goal-driven behavior, and in contrast to model-based reinforcement learning, which heavily relies on the reward design to evoke goal-driven behavior, as its dynamics represent what is possible, rather than what is desirable. Additionally, it is easy to design goal likelihoods when goals provide a significant amount of information that obviates the need to do any manual tuning. The main assumption is that one of the goals in the goal set is reachable within the time-horizon of the model (as judged by the prior). When this assumption is satisfied, we can construct hyperparameter-free distributions like (A), (B), and (C) in the paper. (A) simply constrains the final state of the plan to land at one of the states provided by the outer planner, requires no hyperparameters, and outperforms every method that isn’t ours in all settings. This is analogous to a boundary value problem. If the states in (A) are waypoints along a route, then (B) is preferable: it allows for very easily choosing positions between waypoints.
>
> > “Could the authors please include a variance analysis (using different seeds) in Tables 3 and 4? Previous papers have reported high variance in similar settings (cf., Codevilla et al 2019), and this is a common issue in IL/RL.”
>
> We agree that a variance analysis will strengthen our experimental findings, and are performing the recommended seed-randomization sweeps for each experiment (with three seeds, following Codevilla et al 2019). Due to their computational burden, not all of these experiments will have completed before the response period ends. In light of this, we have prioritized these experiments by focusing on (1) our recommended goal likelihoods for this task (2) the most difficult experimental settings. For (1), we recommend the constraint-based likelihoods due to their absence of hyperparameters. For (2), we primarily evaluate in the dynamic world on the held-out scene (Town02). Here we report the results of this analysis, and have updated the paper with these results. Means and their standard errors are reported. Overall, these results agree with our originally reported results.
>
> “Region-Final St. Indicator S.” on Town02 (test) are: Success: 88% ± 3.3,  Red light: 2.57% ± 0.04, Wrong-lane: 0.49% ± 0.32, Off-road: 2.6% ± 1.1
> “Region-Final St. Indicator S.” on Town01 (train) are: Success: 96% ± 1.9, Red light: 0.89% ± 0.43, Wrong-lane: 0.05% ± 0.01, Off-road: 0.11% ± 0.01.
> “Line Segment Final-St. Indicator” on Town02 (test) are: Success: 88% ± 3.3, Red light: 34.8% ± 2.4, Wrong-lane: 0.15% ± 0.04, Off-road: 1.79% ± 0.34

---

> > ### Author Response · Authors · 2019-11-12
> > **Response to Review 3 (part 2)**
> >
> > > “How important is knowing $\lambda$ (traffic light state) perfectly in practice? Can the robustness to noise in $\lambda$ be experimentally assessed? I would also clarify in section 4 and Table 2 that other methods do not use $\lambda$ (the traffic light state), which is a signal very strongly correlated with the "ran red light" metric.”
> >
> > We agree this is an interesting question to investigate and thank the reviewer for suggesting this additional robustness experiment. To perform it, we simulated noise in $\lambda$ by flipping the light state with 20% probability, corresponding to a light state detector that has 80% accuracy on average. Flipping means that if the light is green, then changing $\lambda$ to indicate red, and if the light is red, then changing $\lambda$ to indicate green. We performed this following the experimental method of “Region Final-St. Indicator S.” in dynamic Town02, and ran it with three separate seeds. The means and their standard errors are:
> > Success: 76% ± 5.0, Red light: 34.8% ± 2.4, Wrong-lane: 0.15% ± 0.04, Off-road: 1.79% ± 0.34. Relative the above reported results that did not model $\lambda$ noise — (Success: 88%± 3.3,  Red light: 2.57% ± 0.04, Wrong-lane: 0.49% ± 0.32, Off-road: 2.6% ± 1.1.) — the conclusion we draw is that the approach can still achieve success most of the time, although it tends to violate red-lights more often. Qualitatively, we observed the resulting behavior near intersections to sometimes be “jerky”, with the model alternating between stopping and non-stopping plans. We hypothesize that the model itself could be made more robust if the noise in $\lambda$ was also present in the training data. We have added the results of this experiment to the existing robustness experiments section in Appendix E.
> >
> > We have also clarified the traffic light presence in Sec 4 and Table 2.
> > > “More generally, what is the robustness of this approach to uncertainty / noise in \phi? Although it is typically available (as the authors mentioned) it is never perfect in practice. Can this be handled in a principled probabilistic way as an extension of the current formulation?”
> >
> > We agree that this is also an interesting general question to investigate, because the existence of both (1) observation noise and (2) uncertain/out-of-distribution observations is indeed an important practical issue for autonomous vehicles. Although our current method only conditions on our current observation, several extensions could help mitigate the negative effects of both (1) and (2). For (1), a Bayesian filtering formulation is arguably most ideal, to better estimate (and track) the location of static and dynamic obstacles under noise. However, such high-dimensional filtering are often intractable, and might necessitate approximate Bayesian deep learning techniques, RNNs, or frame stacking, to benefit from multiple observations.  Addressing (2) would ideally be done by placing a prior over our neural network weights, to derive some measure of confidence in our density estimation of how expert each plan is, such that unfamiliar scenes generate large uncertainty on our density estimate that we could detect, and react cautiously (pessimistically) to. One way to address the situation if the distributions are very different is to adopt an ensembling approach [A] in order for the method to determine when the inputs are out of distribution — the ensemble will usually have higher variance (i.e. disagree) when each element of the ensemble is provided with an out-of-distribution input. For instance, this variance could be used as a penalization in the planning criterion. We updated the paper to briefly mention this in the Discussion section and include the full discussion in the appendix.
> >
> > [A] Lakshminarayanan et al. Simple and scalable predictive uncertainty estimation using deep ensembles. NeurIPS 2017.
> >
> > > “The current model does not factor the influence of the agent on its environment. Is this framework limited to open loop planning, or does this open interesting future research directions towards closing the loop? It seems to be a key open problem to at least discuss in Section 5.”
> >
> > We agree that extending the method to closed-loop settings is an interesting future direction, for instance by using a model that reasons about the interactions between agents when planning the controls of the ego-agent. We included mention of this direction in Sec 5.
> >
> > > ”Additional Feedback: Figure 5 is confusing..typos in Appendix”
> >
> > We agree that Figure 5 is somewhat tangential, but were torn because we thought some readers might benefit from its presence. We have compromised by moving it to the appendix near the architecture table, and merely refer to it in Sec 2.4. Thank you for finding these typos, we have fixed them.

---

### Official Review · AnonReviewer1 · 2019-10-22
**Official Blind Review #1**

**Rating:** 6

**Review:**

The paper propose imitative models that learns goal-based probabilistic models of expert demonstrations, and use this to perform test-time planning and control of certain goal-directed behavior. The paper demonstrates extensive and impressive experiments on the CARLA simulator and outperforms existing approaches in the success metric.

The idea is quite simple: given a set of states, learn a probabilistic model that assigns high probability likelihood to expert behavior. After training, inference is performed to optimize the likelihood of a goal according to an expert prior. The paper discusses extensively how goal-based likelihood functions would be designed for autonomous driving, and the architectures for good q(s|\phi) in detail, which are of engineering importance in self-driving applications. While I believe the method described might not be significantly novel technically, I believe the paper made nice contributions in terms of the autonomous driving application.

Minor comments:
	- It seems like the term "state" is used to represent the agent's location on the ground plane, which is also not technically the entire state information?
	- Is the argmax in (1) a strict equality? I guess you would assume q(s|\phi) = p(s|\phi) for this to be always true?
	- How many iterations does it take to converge in Algorithm 2? It seems you would need to updates z{1:T} for all newly encountered goals?


**Experience Assessment:**

I do not know much about this area.

**Review Assessment: Checking Correctness Of Derivations And Theory:**

I carefully checked the derivations and theory.

**Review Assessment: Checking Correctness Of Experiments:**

I assessed the sensibility of the experiments.

**Review Assessment: Thoroughness In Paper Reading:**

I read the paper at least twice and used my best judgement in assessing the paper.

---

> ### Author Response · Authors · 2019-11-12
> **Response to Review 1**
>
> We thank R1 for their insights, constructive feedback, and favorable impression of our method’s contribution to the application.
>
> > “It seems like the term "state" is used to represent the agent's location on the ground plane, which is also not technically the entire state information?”
>
> Yes, you are right, it is not enough information to construct a Markovian world state. Technically, the model is a Partially-Observed Markov Decision Process (POMDP), however, we left out this discussion and formalization because it is not needed for the purposes of the paper. In order to incorporate your point and enhance clarity, we included a specific mention in Sec 2.3 that the system is best characterized as a POMDP, but that we call observations of position as “states” for brevity, because the POMDP formalization is not needed for the exposition of our approach.
>
> > “ Is the argmax in (1) a strict equality? I guess you would assume $q(s|\phi) = p(s|\phi)$ for this to be always true?”
>
> It is a strict equality when we take $p(s|\phi)=q(s|\phi)$, yes. This corresponds to a posterior with prior $q(s|\phi)$.
>
> > “How many iterations does it take to converge in Algorithm 2? It seems you would need to updates z{1:T} for all newly encountered goals? “
>
> Yes, z must be replanned when new goals are encountered (The goals in Alg 1 Line 3 generally vary). In practice, we performed optimization over a batch of 120 independent z vectors, applying gradient descent in parallel, and using the z of the best plan (under the planning criterion) as the result. With this approach, we observed Alg 2 to usually converge quickly, and therefore instead of specifying a specific convergence criterion, we ran 10 steps of gradient descent. We included these important details in Appendix A.1.
>
> > “I believe the method described might not be significantly novel technically”
>
> Our method is a new principled technical framework that combines IL and MBRL. The result is an approach that accrues many of the benefits of Imitation Learning and Model-Based Reinforcement Learning while simultaneously mitigating their downsides. Historically, it has been difficult to use Imitation Learning (IL) to learn agents that are easily directable to new goals; Model-Based Reinforcement Learning (MBRL) requires significant reward engineering to work well. Our approach easily integrates test-time goal-direction procedures to direct our learned agent to new goals, and requires significantly less reward engineering than Model-Based Reinforcement Learning because the learned agent already internally represents and seeks to perform desirable behavior. We found our approach to achieve very good performance in the CARLA simulator for a wide variety of different goal-direction procedures.

---

### Official Review · AnonReviewer2 · 2019-10-22
**Official Blind Review #2**

**Rating:** 8

**Review:**

The paper takes a model-based approach to "imitation learning". It first learns a (flow) "model" to assign likelihoods to trajectories being expert-like as well as sample expert-like trajectories. This is then combined with an estimate for trajectories being certain goal conditioned, where the goals come from a route planner. Results are shown in the context of autonomous driving in the CARLA simulator with a PID controller tracking the open-loop plan from the planner.

One major issue with the paper is that all the main contribution seem to be in the appendix. If I were to summarize it, the paper introduces an interesting way of integrating two different experts to perform learning from demonstration. On the one hand, you have the A* algorithm as an expert providing what waypoints to follow. On the other hand, you have the expert demonstrating how to drive around on the road. The question becomes, how do we reconcile these "experts" acting at different abstraction levels so that our system works in a more general setting. This kind of a setting is definitely not as general as the paper tries to make it out to be, but nevertheless it's reasonably broad and useful.
The idea to of different subsystems for route planning and path planning are not new, but the way it's done here does seem interesting to me.

Note that Sec 3 related work CILS description seems hard to understand. Moreover, the claim "MBRL can also plan, but with a one-step predictive model of possible dynamics", seems incorrect. One can do multi-step predictions and use those to do model-based RL as well. So it's unfair to claim that this other approach is MBRL and yours is not on that basis.

Experiments are interesting although quite dense. It's unclear what the initial state distribution look like for these experiments though. I'm guessing CARLA is otherwise a deterministic simulator?

Now I am going to do something that you are not supposed to do as a reviewer. Tell the authors the paper they should have written, rather than the paper that was submitted. Apologies for that.
The terminology of Inverse RL, Imitation Learning and Apprenticeship Learning is a mess in the literature unfortunately and can't blame you not trying to fix it in this paper. However the community would benefit if don't overload these terms. Although all of these fall under learning from demonstrations, it's useful to restrict Inverse RL as trying to figure out the reward function being optimized by an expert, imitation learning as learning the exact behavior of an expert (and therefore not doing better than the expert which are by definition optimal or being applicable to doing something other than the expert) while apprenticeship learning as learning from demonstrations to perform even better at the task even generalizing to other things than the demonstrations. With these definitions under the belt the work in this paper is better positioned as about apprenticeship learning. Learning a model from the expert demonstration about how the world works and then using the same for accomplishing different tasks for which there are _no demonstrations_ doesn't sound like imitation?

**Experience Assessment:**

I have published one or two papers in this area.

**Review Assessment: Checking Correctness Of Derivations And Theory:**

I assessed the sensibility of the derivations and theory.

**Review Assessment: Checking Correctness Of Experiments:**

I assessed the sensibility of the experiments.

**Review Assessment: Thoroughness In Paper Reading:**

I read the paper thoroughly.

---

> ### Author Response · Authors · 2019-11-12
> **Response to Review 2**
>
> We thank R2 for their insights, constructive feedback, and summary of our method’s contribution.
>
> > “One major issue with the paper is that all the main contribution seem to be in the appendix. If I were to summarize it, the paper introduces an interesting way of integrating two different experts to perform learning from demonstration.”
>
> We agree that the paper can also be viewed as a way to integrate experts at different levels of abstraction. In this view, different goal likelihoods represent different ways for the outer expert to communicate tasks to the inner learned controller. To convey this important perspective, we  moved Algs 1 and 2 into Sec 2.4 in the main body, and add a paragraph of discussion regarding this perspective in Sec 2.1.
>
> > “The claim "MBRL can also plan, but with a one-step predictive model of possible dynamics", seems incorrect. One can do multi-step predictions and use those to do model-based RL as well. So it's unfair to claim that this other approach is MBRL and yours is not on that basis.”
>
> We agree, so we adjusted the text to remove the incorrect assertion that MBRL is always used with a one-step model. Instead that sentence reads “MBRL can also plan with a predictive model, but its model only represents possible dynamics.”
>
> > “It's unclear what the initial state distribution look like for these experiments”
>
> The initial states are sampled from a finite set of starting locations provided by the CARLA simulator. We have clarified this in Section 4.
>
> > “The terminology of Inverse RL, Imitation Learning and Apprenticeship Learning is a mess in the literature unfortunately and can't blame you not trying to fix it in this paper... it's useful to restrict Inverse RL as trying to figure out the reward function being optimized by an expert, imitation learning as learning the exact behavior of an expert...apprenticeship learning as learning from demonstrations to perform even better at the task even generalizing to other things than the demonstration...Learning a model from the expert demonstration about how the world works and then using the same for accomplishing different tasks for which there are _no demonstrations_ doesn't sound like imitation?”
>
> We agree that the scattered terminology is difficult to rectify in a single paper, and that our approach can be described as apprenticeship learning. We also agree that our approach is not perfectly categorized as either Imitation Learning or Model-Based Reinforcement Learning — in Sec 5 we said “Learning an Imitative Model resembles offline IL” and “Inference with an Imitative Model resembles trajectory optimization in MBRL, enabling it to both incorporate new goals and plan to them at test-time, which IL cannot”. We have since made this point clearer by adding it to Sec 2.1.
>
> The view of our approach as apprenticeship learning is connected to your first point that the approach is a way of integrating two different levels of controllers. The apprentice is learned at a mid-level, and is amenable to different ways of providing high-level direction. We thus integrated combined discussion of the apprenticeship learning and mid-level/high-level controller perspective into Section 2.1. However, we are not aware of the term “apprenticeship learning” being used to mean something distinct from “imitation learning”. While we could define “apprenticeship learning” as suggested, this definition would be strengthened if we included references that use “apprenticeship learning” to mean something besides “imitation learning”; we are not aware of any such references.

---

> > ### Comment · AnonReviewer2 · 2019-11-14
> > **Re: apprenticeship learning**
> >
> > > we are not aware of any such references
> >
> > Why not _be_ that reference?
> > Anyway, it's unfortunate the original apprenticeship learning papers ended up just doing imitation learning and the term is likely beyond saving. I like how you use "apprentice" though!

---

> > ### Comment · AnonReviewer2 · 2019-11-14
> > **Re: Updated scores**
> >
> > I'm leaning towards accept. Hopefully the source code would be released to help reproducibility? I understand there are a lot of moving parts and you have shared the main parts.

---

> > > ### Author Response · Authors · 2019-11-15
> > > **Source code**
> > >
> > > We strongly agree that a source code release will help others reproduce and build upon our results, and are actively working to make it easy to use in order to maximize its benefit to the community when we release it.

---

### Decision · Program_Chairs · 2019-12-19

**Decision:**

Accept (Poster)

**Comment:**

This paper proposes to build an 'imitative model' to improve the performance for imitation learning. The main idea is to combine the model-based RL type of work to the imitation learning approach. The model is trained using a probabilistic method and can help the agent imitate goals that were previously not easy to achieve with previous works.

Reviewers 2 and 3 strongly agree that the paper should be accepted. R3 has increased their score after the rebuttal, and the authors' response helped in this case. Based on reviewers score, I recommend to accept this paper.